# Dynamic character displacement among a pair of bacterial phyllosphere commensals in situ

Lucas Hemmerle[1], Benjamin A. Maier[1], Miriam Bortfeld-Miller[1], Birgitta Ryback[1], Christoph G. Gäbelein[1], Martin Ackermann[2,3] & Julia A. Vorholt [1✉]

Differences between species promote stable coexistence in a resource-limited environment. These differences can result from interspecies competition leading to character shifts, a process referred to as character displacement. While character displacement is often interpreted as a consequence of genetically fixed trait differences between species, it can also be mediated by phenotypic plasticity in response to the presence of another species. Here, we test whether phenotypic plasticity leads to a shift in proteome allocation during co-occurrence of two bacterial species from the abundant, leaf-colonizing families *Sphingomonadaceae* and *Rhizobiaceae* in their natural habitat. Upon mono-colonizing of the phyllosphere, both species exhibit specific and shared protein functions indicating a niche overlap. During co-colonization, quantitative differences in the protein repertoire of both bacterial populations occur as a result of bacterial coexistence *in planta*. Specifically, the *Sphingomonas* strain produces enzymes for the metabolization of xylan, while the *Rhizobium* strain reprograms its metabolism to beta-oxidation of fatty acids fueled via the glyoxylate cycle and adapts its biotin acquisition. We demonstrate the conditional relevance of cross-species facilitation by mutagenesis leading to loss of fitness in competition *in planta*. Our results show that dynamic character displacement and niche facilitation mediated by phenotypic plasticity can contribute to species coexistence.

[1] Institute of Microbiology, ETH Zurich, Zurich, Switzerland. [2] Department of Environmental Systems Science, ETH Zurich, Zurich, Switzerland. [3] Department of Environmental Microbiology, Eawag, Dubendorf, Switzerland. ✉email: jvorholt@ethz.ch

The mechanisms that generate and maintain genetic and phenotypic variation in natural populations are central to ecology and evolution. Character displacement, the evolutionary divergence of competing species, plays a fundamental role in the assembly of diverse communities[1,2]. Sympatric species compete for the same set of limited resources, and thus natural selection might favor diversification of their use. As a result, phenotypes may emerge in which alternative resources are exploited, thereby reducing competition for nutrients[3–6]. Once separated in niche space, a species may modify the environment and directly or indirectly enhance its own growth and survival[7,8]. As a result, competitive interactions are altered among species through mechanisms such as increased access to limiting resources, which in turn has impacts on biodiversity and prevalence[9].

The nature of the interactions that generate patterns of diversity remains difficult to examine[10]. To provide a mechanistic basis, evolution experiments and evolutionary analysis of bacterial populations have been used to study the processes that lead to diversifications, and these model studies have illustrated how competitive and facilitative interactions between distinct genotypes can evolve and contribute to their coexistence[8,11–15]. These and other studies have added to our understanding of microbial community assembly by providing exemplary analysis of the evolutionary processes that drive diversity.

A complementary and non-exclusive perspective on the interaction of microbial species that may contribute to coexistence is their potential to dynamically shift their phenotype in response to the presence of other species. This process is referred to as phenotypic plasticity and has been underappreciated in mediating character displacement[16–18]. According to this idea, when two extant species encounter each other and compete for limited resources, they diversify through changes in gene expression and thus become less similar (Fig. 1). The latter can be seen as the result of an ecological interaction, but may be the consequence of an evolutionary process in the past. Such phenotypic diversification could be exhibited by competing species to reduce their niche overlap. In fact, it has been hypothesized that phenotypic plasticity may precede genetic differences among species[18].

While measuring differences among species is more straightforward, discerning the processes causing character displacement is inherently difficult[2,10] and requires empirical quantifiable differences with appropriate controls. Here, we set out to test the phenotypic plasticity of two commensal species and to examine whether they become phenotypically more dissimilar when co-occurring compared to being present alone. To provide context for their interaction, we chose two bacterial strains isolated from the phyllosphere of the model plant Arabidopsis and tested their interaction *in planta*. The phyllosphere, the above-ground parts of plants dominated by leaves, is an important ecosystem due to its contribution to carbon dioxide fixation in terrestrial systems and indigenous bacteria may impact plant growth and health[19–21]. In addition, the phyllosphere is also a highly suitable habitat to study microbial interactions because it represents a discrete habitat or, more precisely, a sum of discrete habitats[22–24] and allows quantifying population sizes in a well-defined area. The phyllosphere, a heterogeneously structured and oligotrophic environment, has been shown to maintain high and reproducible microbial biodiversity[25,26]. As a result, due to limited resources, it provides the ground for species interactions and opportunities to discover rules for community assembly in this habitat[27,28].

Culture-independent studies on the composition of the phyllosphere microbiota of different plant species revealed that reoccurring phylogenetic structures can be observed[29–33]. Bacteria belonging to the phylum Proteobacteria are most commonly found,

followed by Actinobacteria, Bacteroidetes, and Firmicutes across distinct plant species[26,29,34]. These bacteria are mostly found on the surface of leaves, where they can form aggregates[20,35–39]. These bacterial aggregates can be constituted by a single bacterial species or by several species and are of great interest for the elucidation of bacteria-bacteria interactions[40,41].

Several studies have addressed the physiological adaptation strategies involved in leaf colonization by individual bacterial species of commensals[42–45] and plant pathogens[46–48]. Complementary studies have provided the first insights into the physiological capacity of the coexisting phyllosphere microbiota *in planta* using metaproteomics[25,49]. While providing a snapshot of the state of microbial communities in situ, the dynamic processes underlying niche occupancy as a consequence of species competition are currently not well understood. Because interspecies interactions in situ are complicated by the presence of a multitude of species, simplified systems are needed, that allow to dissect dynamic behavior of populations.

In the present study, we use an established gnotobiotic *Arabidopsis thaliana* model system[50,51] to test whether two species undergo shifts in the phenotype upon co-colonization. We select two strains representing the most common colonizing proteobacterial families *Sphingomonadaceae* and *Rhizobiaceae* and apply mass spectrometry-based proteomics to quantify proteome-level changes in each strain upon co-colonization of plants

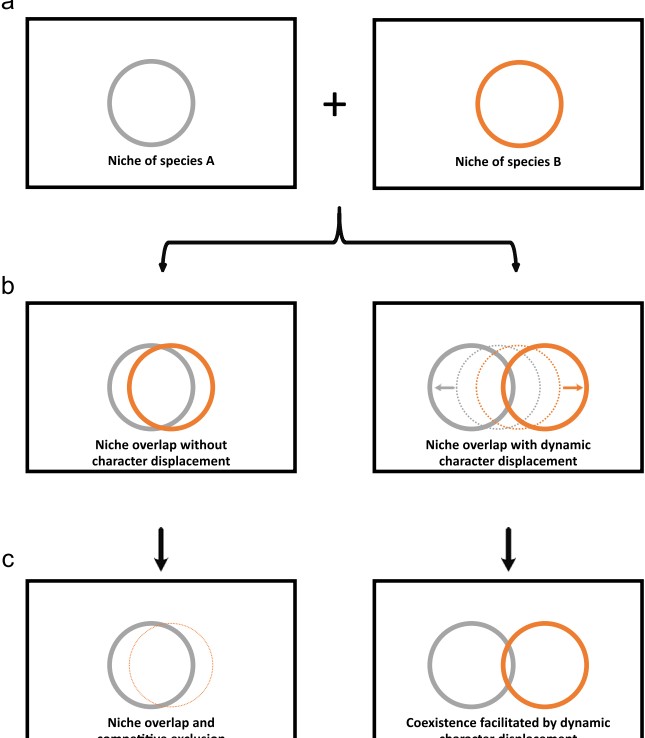

**Fig. 1 Conceptional overview of niche occupancy by bacteria colonizing the plant in mono-association and upon coexistence. a** General niche of two species, A (gray) and B (orange), in case they do not encounter each other in a given habitat. **b** During co-occurrence, the niches of the two species overlap substantially, leading to competition for shared resources (left) but also to potential for a dynamic character displacement (right, indicated by arrows). **c** As a consequence, one species may displace the other species from the niche resulting in competitive exclusion (left). Alternatively, a dynamic character displacement may occur, reducing the original niche overlap and leading to coexistence (right). The size of the circle indicates productivity in the ecological niche. In this study, niche occupancy was measured using a proteomics approach.

compared to mono-colonization. In addition, we identify the specific adaption of each strain to the *in planta* condition compared to growth on artificial media. As expected, we confirm that both species are generalist bacteria that have the potential to utilize a variety of substrates such as glucose or sucrose as an available, yet limiting carbon source on leaves[52]. In addition to facilitation, we observe a dynamic character displacement that might contribute to the coexistence of the populations. We confirm relevant traits by mutagenesis and testing *in planta*.

## Results

### Assessment of the interaction of two phyllosphere strains based on population sizes and proteome allocation *in planta*.
To study the interaction between phyllosphere microbiota strains, we selected two representative strains from the Alphaproteobacteria, the most abundant bacterial class found on *A. thaliana* leaves[26]. Among Alphaproteobacteria the bacterial families of *Sphingomonadaceae* and *Rhizobiaceaea* are the most common and include strains with plant beneficial functions[20,21,29,50,53]. Previous experiments have shown that the genera of *Sphingomonas* and *Rhizobium* have the potential to affect community assembly and structure[27]. Furthermore, both genera have overlapping metabolic capacities and occur in the spatially defined, oligotrophic environment of the *A. thaliana* phyllosphere[26,29]. Specifically, we chose *Sphingomonas* Leaf257 and *Rhizobium* Leaf68 and confirmed their consistent colonization capacities in mono-colonization using a gnotobiotic plant growth system (see Methods). Both strains were isolated from leaf washes and colonize leaves as epiphytes[29,54]. We also ensured that both strains colonized the phyllosphere upon co-colonization. In addition, we confirmed their co-occurrence during colonization at the microscale (Supplementary Fig. 1).

We then conducted four independent large-scale colonization experiments in which we inoculated the plants with either *Rhizobium* Leaf68 and *Sphingomonas* Leaf257 or a mixture of both. In addition, we determined the bacterial population sizes upon mono- and co-colonization. Both strains colonized the phyllosphere, with *Sphingomonas* Leaf257 reaching $1.6 \times 10^9$ CFUs per gram fresh weight (CFUs g$^{-1}$ FW) (Fig. 2a) and *Rhizobium* Leaf68 of $2.6 \times 10^8$ CFUs g$^{-1}$ FW (Fig. 2b). During co-colonization of the phyllosphere, both strains significantly changed their colonization capacity (Kruskal–Wallis, *P*-value <0.01). *Sphingomonas* Leaf257 was reduced to $7.5 \times 10^8$ (0.46-fold) (Fig. 2a) and *Rhizobium* Leaf68 increased to $1.2 \times 10^9$ CFUs g$^{-1}$ FW (4.6-fold) (Fig. 2b). The population sizes indicated an interaction between the two strains during co-colonization in the phyllosphere, as both populations changed in number relative to mono-colonization. Because commensal bacteria can elicit plant responses[55] and may indirectly affect colonization of other bacteria, we tested the interaction of both strains using a selection of plant immunity mutants (Methods). We found that the bacterial interaction was robust in different plant backgrounds (Supplementary Fig. 2).

We determined the proteomes of the bacterial populations *in planta* using a washing protocol (Methods) (i.e., *Rhizobium* Leaf68, *Sphingomonas* Leaf257 and both upon co-colonization) from the corresponding experiments enumerated above from four biological replicates. For protein determination, we used liquid chromatography-mass spectrometry (LC-MS/MS) operated in data-dependent acquisition mode (Methods, Supplementary Fig. 3a). Notably, to improve protein quantification and to compensate effects of sample complexity of bacteria during mono-colonization compared to co-colonization, we also measured an artificial mixture from bacteria grown in mono-colonization *in planta* after cell lysis and protein digestion. In addition to the biological samples of bacteria grown *in planta*, we

harvested the two strains grown each on solidified artificial medium on agar plates incubated under the same light and temperature regime in plant growth chambers in four independent replicates. Our MS-based approach allowed the detection of up to 2500 proteins per strain with normalized protein abundances spanning more than six orders of magnitude (Supplementary Fig. 3b–d) and the quantification of about 2000 proteins detected with two or more unique peptides for each strain. We identified proteins that were induced or reduced when the bacteria encountered each other *in planta*, confirming that both bacteria interacted with each other (Fig. 2c, d), as already suggested by the population sizes (Fig. 2a, b).

### Individualized and common adaptation strategies to colonize the phyllosphere.
Before investigating the interaction between both bacteria, we first analyzed the physiological adaptation to the *in planta* conditions of each strain compared to growth on artificial media using proteomics. For *Sphingomonas* Leaf257, we identified 184 proteins that were induced *in planta* (Supplementary Fig. 4a, Supplementary Data 1). Among these, a number of proteins indicate an adaptation to environmental stress in terms of desiccation, oxidative stress and light in line with earlier reports on leaf strains[25,44]. Exemplarily, these included catalase counteracting oxidative stress as well as deoxyribodipyrimidine photolyase-related protein (COG3046) and UmuC as SOS induced DNA polymerase enabling DNA damage repair. We also identified a glycosyltransferase (ASF14_08960) as a highly induced protein (Log$_2$ fold-change >35) and its paralog (ASF14_03660, >4-fold) that was also induced upon plant colonization in *Sphingomonas melonis* Fr1, another phyllosphere commensal[44]. Glycosyltransferases might be involved in lipopolysaccharide and biofilm synthesis[56–58] and their strong induction might highlight their relevance in phyllosphere colonization. In addition, fourteen distinct TonB-dependent receptors were induced upon phyllosphere colonization that might be involved in uptake of iron, carbohydrates or amino acids[25,44].

For *Rhizobium* Leaf68, we found 356 proteins induced *in planta* (Supplementary Fig. 4b, Supplementary Data 2). The most strongly induced protein (about 1000-fold) *in planta* was an alcohol dehydrogenase (ASF03_19755), closely followed by several hypothetical proteins. We also identified two proteins that are involved in the detoxification of reactive oxygen species, a peroxidases (ASF03_20400) and a superoxide dismutase (ASF03_03695) with fold changes higher than 10 indicating that the bacterium has to cope with ROS-stress during phyllosphere colonization[25,42]. In addition, we found a predicted biotin sulfoxide reductase (ASF03_18405, 5.5-fold) and a biotin synthase (ASF03_03440, 5.2-fold) to be induced, which may suggest the need of *Rhizobium* Leaf68 to scavenge a biotin derivative. Two proteins (ASF03_19470 and ASF03_19480) involved in the synthesis of succinoglycan, an exopolysaccharide, were also strongly induced with fold changes of more than an order of magnitude (Supplementary Data 2). Succinoglycans are exopolysaccharides (EPS) known for their thermal stability, high water retention and viscosifying properties in many soil bacteria, including *Rhizobia*[59–61]. EPS production is a major factor in the formation of bacterial aggregates and might therefore benefit *Rhizobium* Leaf68 to colonize and form aggregates in the phyllosphere[26,36]. Another phyllosphere-specific response of *Rhizobium* Leaf68 was the induction of two sulfonate mono-oxygenases (ASF03_18445, 20.6-fold; ASF03_18375, 5.4-fold) and several ABC transporter binding proteins (CysA, 7.5-fold; CysP, 13.5-fold; SsuA, 32-fold; SsuB, 27-fold), proteins involved in sulfur acquisition during sulfur limiting conditions[62]. Furthermore, we found the two subunits of sulfate adenylyltransferase

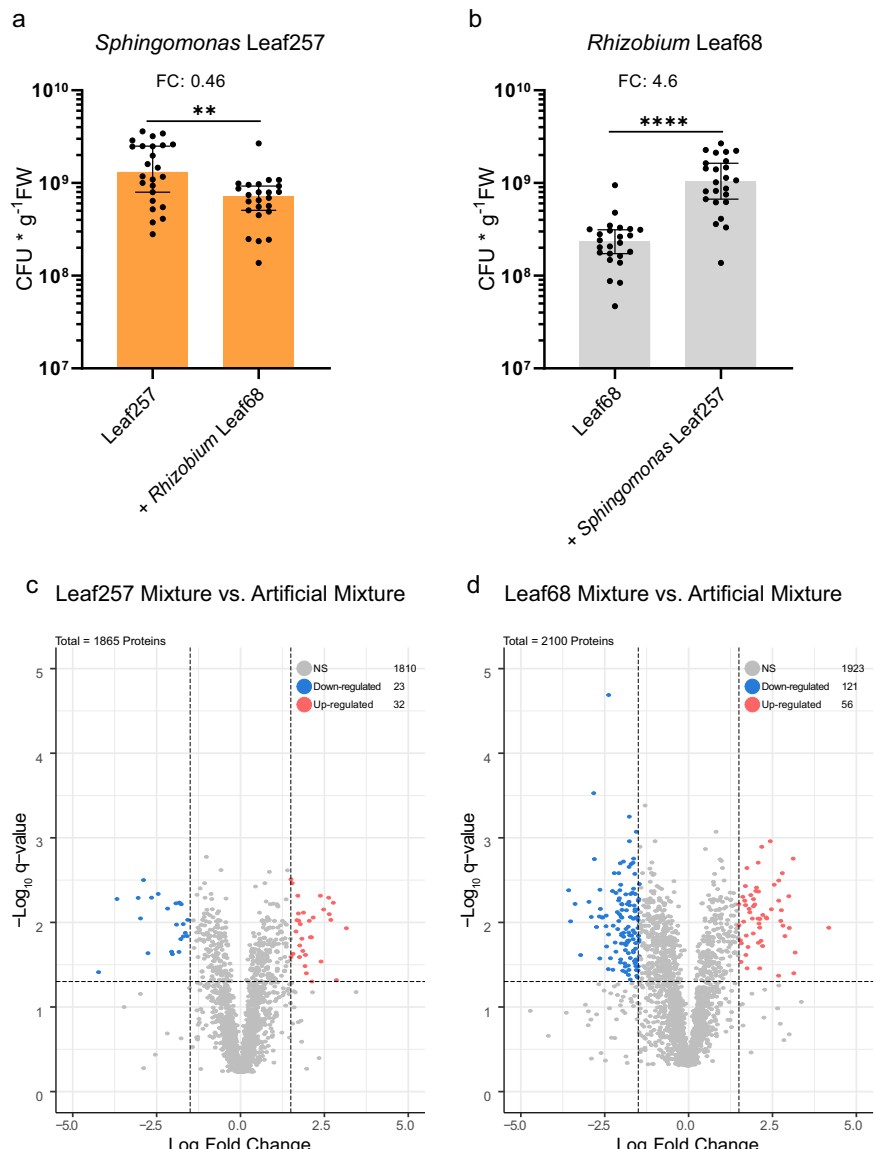

**Fig. 2 Changes in the colonization potential and proteomes upon co-colonization of *Arabidopsis thaliana*. a, b** Cell numbers (CFUs per gram fresh weight) of *Sphingomonas* Leaf257 (orange), or *Rhizobium* Leaf68 (gray) alone and upon co-colonization harvested after 11 dpi. Average fold-change (FC) between the two conditions is indicated above the bar. Graphs depict the combined data from four independent biological replicates with each dot representing the CFUs obtained from a single plant with six plants per biological replicate ($n = 24$). In the graphs, the median and 95% confidence interval are shown. Statistical analysis was performed using the Kruskal–Wallis test (P-values: *p-value < 0.05, **p-value < 0.01, ***p-value < 0.001, and ****p-value < 0.0001). Proteome changes are depicted as volcano plots for (**c**) Leaf257 during co-colonization of the plant with Leaf68 and (**d**) Leaf68 during co-colonization of the plant with Leaf257.) Proteomics data were obtained from four independent biological replicates ($n = 4$) (see methods). The adjusted P-value (q-value) cutoff was set to 0.05 (horizontal line) and Log$_2$ Fold change cutoffs are set to >1.5 or < −1.5 (vertical lines). In the graphs, only proteins detected with at least two unique peptides are shown. The total number of proteins, not significantly regulated (NS), and Down- / Up-regulated are indicated in each graph. For (**c**, **d**) statistical analysis was performed using one-way ANOVA and resulting P-values were corrected for multiple comparisons using Benjamini–Hochberg. Source data for (**a**, **b**) are provided as a Source Data file.

(ASF03_03950, >20-fold; ASF03_03955, 5.4-fold), an enzyme involved in sulfate assimilation, significantly induced, supporting the hypothesis that sulfur is limiting in the phyllosphere congruent with earlier studies[44,63,64].

The induced proteins upon plant colonization in mono-colonization indicated that unique but also shared adaptation might occur. To compare the induced proteome systematically, we analyzed the data for orthologous groups (OGs) of proteins and determined which OGs were significantly regulated and shared between both strains (q-value < 0.05 and Log$_2$ fold-change >1.5 or <−1.5). From a total of 62 shared OGs, 24 were induced

and 38 reduced when comparing *in planta* regulated proteins compared to in vitro conditions (Supplementary Fig. 4c, Supplementary Data 3). Next, we grouped all OGs by their functional category to see which functions are shared between both bacteria during phyllosphere colonization (Supplementary Fig. 4d, Supplementary Data 3). Some of the most induced shared protein orthologues are two glycosyltransferases (COG0438, COG0463), proteins involved in the molybdenum cofactor and thiamine biosynthesis, catalases, and one of the key enzymes involved in the glyoxylate cycle, isocitrate lyase (COG2224). The other key enzyme in the glyoxylate cycle, malate synthase

(COG2225), was also induced in both strains but only 1.3-fold in *Rhizobium* Leaf68. The induction of the glyoxylate cycle upon plant colonization was previously observed in another phyllosphere commensal, *S. melonis* Fr1[44]. Additionally, a pyruvate dehydrogenase (COG0028) was induced in both strains with similar fold changes as well as the histidine kinase sensor protein EnvZ. The EnvZ histidine kinase is involved in the response to changes in osmolarity in *E. coli*[65], a stress that bacteria also face during phyllosphere colonization[66]. Furthermore, we observed a glutathione transferase, an alcohol dehydrogenase, acetolactate and glutamine synthetase, and several outer membrane proteins. Interestingly, D-3-phosphoglycerate dehydrogenase was induced in both strains, which is the committing and rate limiting step in the phosphorylated pathway of L-serine biosynthesis, consistent with findings in *S. melonis* Fr1[44]. Furthermore, both strains downregulated three OGs involved in translation, ribosomal structure and biogenesis, and transcription respectively, which could be due to lower growth rates *in planta* compared to artificial media. Notably, we also found flagellin (COG1344) reduced in line with earlier studies[44,67]. Whether or not this strategy serves to minimize detection by the plant immune system[68,69] remains to be shown. Common OGs downregulated by both strains were a glycerol-3-phosphate dehydrogenase (COG0578), an alpha-L-arabinofuranosidase (COG3534) and several hypothetical proteins of which one is potentially a glycosyl transferase (COG0463).

In summary, the majority of shared regulated OGs were associated with intracellular stress responses as well as membrane-, cell wall- and envelop-biogenesis, indicative of the importance of cell-barrier remodeling in bacterial adaptation to the rather hydrophobic and water depleted environment of the phyllosphere in accordance with previous studies[26,36]. Importantly also, shared induced OG related to nutrient uptake and metabolism suggested common adaptation to the oligotrophic phyllosphere environment and potential niche overlap.

**Proteome changes during co-colonization of the phyllosphere**. Next, we analyzed the dynamic niche occupancy upon coexistence based on the proteomes of the two strains. To do so, we compared the proteomes generated upon co-colonization with the ones determined under mono-colonization conditions (upon artificial mixture, see above). *Sphingomonas* Leaf257 significantly changed 55 proteins upon co-colonization (q-value < 0.05 and $Log_2$ fold-change >1.5 or <−1.5) of which 32 were induced and 23 were reduced (Fig. 2c, Supplementary Data 4). *Rhizobium* Leaf68 significantly altered 177 proteins (q-value < 0.05 and $Log_2$ fold-change >1.5 or <−1.5) of which 56 were induced and 121 reduced (Fig. 2d, Supplementary Data 5). These changes indicate a reciprocal interaction between the two strains in the phyllosphere.

For *Sphingomonas* Leaf257, one of the most induced proteins upon co-colonization with *Rhizobium* Leaf68 was dipeptidyl peptidase-4 that cleaves X-proline dipeptides from the N-terminus of polypeptides (ASF14_01870, 6.4-fold) (Supplementary Data 4)[70]. Proline-rich proteins are a common building block in the glycoproteins found in plant cell walls, where they are used as anchor points for glycan chains[71]. In agreement with this, several polysaccharide-degrading enzymes were also strongly induced, namely a xylosidase/arabinosidase (ASF14_10230, 5.2-fold), a xylan 1,4-beta-xylosidase (ASF14_16825, 5.2-fold), a glucan 1,4-beta-glucosidase (ASF14_10240, 4.2-fold), and another xylanase (ASF14_16480, 3.1-fold) (Fig. 3a, Supplementary Data 4). In addition, the induction (2.4-fold) of an ABC transporter (ASF14_10270) genomically encoded close to a gene for a predicted xylosidase suggested an increased metabolization

of sugar monomers obtained from xylan degradation, highlighting a possible degradation of plant cell walls. We also found 6-phosphogluconolactonase (ASF14_10560) induced (2.3-fold), which is required, for example, to metabolize glucuronic acid, which forms side chains of xylan, thus supporting the notion of a sugar metabolism from xylan. In addition, *Sphingomonas* Leaf257 induced the iron transporter FeoB (2.9-fold) as well as a predicted ferrichrome-iron receptor (2.5-fold) indicating increased iron limitation during co-colonization of the plant under competitive conditions. Interestingly, less strongly but consistently induced were several enzymes involved in the Shikimate pathway (Supplementary Data 4). Isochorismate pyruvate-lyase (ASF14_15595) catalyzes the reaction from isochorismate to pyruvate and salicylic acid, the latter being observed in other bacteria under iron-limiting conditions[72,73]. The most downregulated proteins by *Sphingomonas* Leaf257 under co-colonization conditions compared to mono-association were a Zn-dependent dipeptidase (ASF14_12990, 18-fold), a putative phosphatase (ASF14_09200, 12-fold) and several hypothetical proteins (>7-fold). In addition, a predicted low-affinity phosphate transporter (ASF14_05655, 3.4-fold) and a phosphate ABC transporter substrate-binding protein (ASF14_05925, 2.4-fold) were downregulated upon co-colonization that were induced *in planta* upon mono-colonization (Supplementary Data 1 and 4). Thus, our results suggest that co-colonization with *Rhizobium* Lea68 results in increased iron- and phosphate shortage for *Sphingomonas* Leaf257 and drives the latter to exploit additional carbon sources by metabolizing components of the plant cell wall, most prominently xylan.

For *Rhizobium* Leaf68, one of the most induced proteins upon co-colonization with *Sphingomonas* Leaf257 was an alpha-galactosidase (ASF03_16130, 9-fold). Another alpha-galactosidase (ASF03_16155) was induced 4-fold, and we observed a small but significant induction for an alpha-galactoside-binding protein (ASF03_16135), suggesting adaptation towards the digestion and uptake of alpha-galactosyl moieties that might be derived from polysaccharides of the plant cell wall (Supplementary Data 5)[74,75]. Several hypothetical proteins were also among the most induced proteins. Interestingly, isocitrate lyase (ASF03_04985; 4-fold) and a glycolate dehydrogenase (ASF03_04840; 6-fold) were induced during co-colonization, suggesting an increased flux towards the glyoxylate cycle as an anabolic pathway. Consistent with this finding, we observed an induction of several enzymes involved in the beta-oxidation of fatty acids (ASF03_04360, 7-fold; ASF03_12360, 2.1-fold; ASF03_12365, 3.4-fold; ASF03_20960, 4.5-fold). Several TonB-dependent receptors, RND efflux systems and other transporters were also induced in response to co-colonization, some of which were predicted ABC transporter for glycerol-3-phosphate (ASF03_16050, 6.4-fold; ASF03_21060, 3.9-fold; ASF03_21075, 3.4-fold; ASF03_18705, 3.3-fold; ASF03_18690, 3.2-fold). In line with this observation, we measured a strong reduction (5-fold) of the glycerol-3-phosphate regulon repressor GlpR (ASF03_11310) indicating an adaptation towards the acquisition of glycerol-3-phosphate through de-repression of the underlying regulon. Glycerol-3-phosphate uptake is notable in the context of fatty acid degradation as this may suggest an adaptation of *Rhizobium* Leaf68 towards growth on lipids. Furthermore, two proteins involved in phosphonate utilization were significantly induced (ASF03_14895, 8-fold; ASF03_14885, 5.8-fold), suggestive of phosphate starvation during co-colonization. Further highlighting an association with phosphate starvation, the PhoB transcriptional regulator (ASF03_21155) associated with phosphate and other nutrient stress networks[76] was induced (Fig. 4a). Taken together, this indicated that in response to co-colonization with *Sphingomonas* Leaf257,

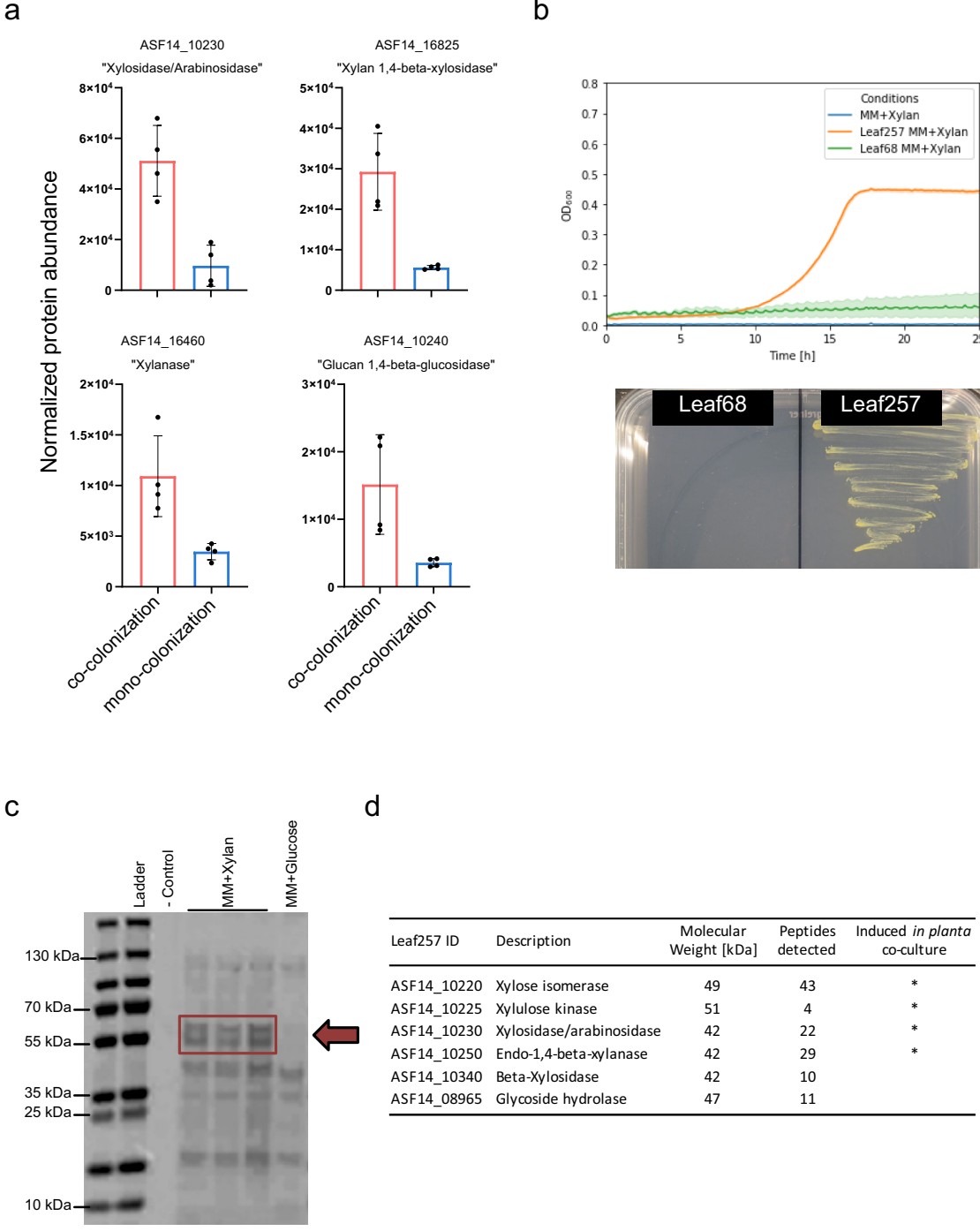

**Fig. 3 Sphingomonas Leaf257 utilizes plant-derived xylan. a** Normalized abundance of proteins related to xylan degradation obtained from the proteomics workflow (see methods, Supplementary Data 4) upon co-colonization (red) and mono-colonization (blue) for four enzymes potentially involved in xylan degradation. Error bars represent the standard deviation. Data are from 4 independent replicates, $n = 4$. **b** Growth curve depicting optical density ($OD_{600}$) of Sphingomonas Leaf257 (orange), Rhizobium Leaf68 (green) and negative control (blue) in minimal medium with 10 mM xylan in liquid (top) and growth on solid minimal medium with 10 mM xylan for Leaf257, Leaf68 (no growth) (bottom). Error bars (top) represent the standard deviation. Data are from 5 independent replicates, $n = 5$. **c** SDS-gel of enriched supernatant (see methods) of Sphingomonas Leaf257 grown in liquid minimal medium with ddH$_2$O (control), xylan or glucose. For MM + xylan three biological independent replicates were prepared. One replicate was prepared for the control MM + glucose. The xylan specific double-band at ca. 55 kDa (red box) was extracted and analyzed using LC-MS. **d** Overview of protein identified in the LC-MS approach. Source data for (**a**, **b**, **c**) are provided as a Source data file.

*Rhizobium* Leaf68 adapted to phosphate scarcity by engaging global regulatory mechanisms. One of the most reduced proteins during co-colonization was alcohol dehydrogenase (ASF03_19755; Log$_2$ fold-change < −6.8), which was striking as the enzyme was highly induced in the mono-colonization

compared to artificial media (Fig. 4a, Supplementary Data 2 and 5). Another observation was that biotin synthase (ASF03_03440), the last enzyme in the biotin biosynthesis, was significantly reduced in *Rhizobium* Leaf68 upon co-colonization with *Sphingomonas* Leaf257.

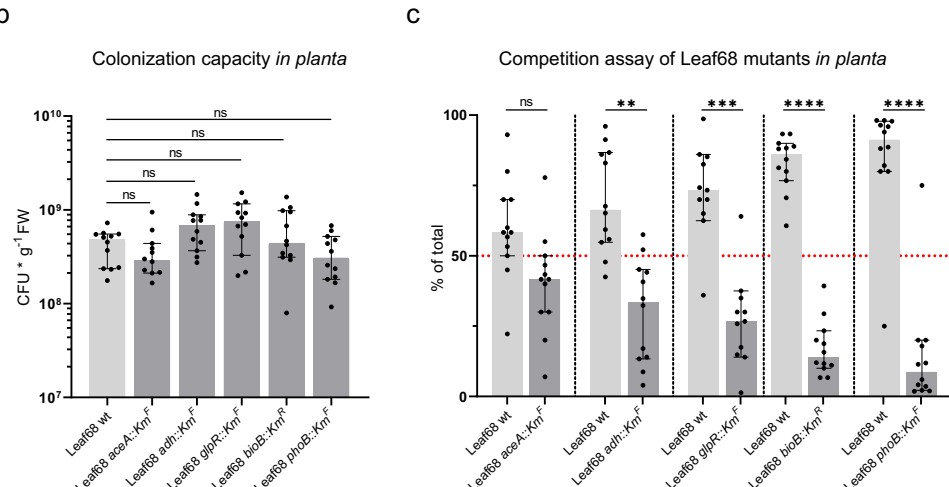

**Fig. 4 Characterization of *Rhizobium* Leaf68 mutants *in planta*. a** Overview table of the five genes of *Rhizobium* Leaf68 selected for mutagenesis and fold-changes of the encoded proteins over the different proteomics conditions (Supplementary Data 2 and 5). *P*-values were calculated using a one-way ANOVA and adjusted for multiple comparisons using Benjamini-Hochberg. **b** Colonization capacity of each (CFUs per gram fresh weight), Leaf68 wild type (light gray), mutants (dark gray) alone *in planta*. Statistical analysis was performed using Kruskal–Wallis test. **c** Competition assay depicting the percentage (%) of total population wild type versus each mutant respectively. A 1:1 mixture is indicated at 50% with a red dotted line. For statistical analysis a two-sided paired t-test was performed (*P*-values: *$p$-value < 0.05, **$p$-value < 0.01, ***$p$-value < 0.001, and ****$p$-value < 0.0001). In graphs b and c, the median and 95% confidence interval are shown. For both graphs each data point represents the values obtained from a single plant with a total of 12 plants for these experiments. Source data for (**b**, **c**) are provided as a Source Data file.

In summary, our results suggest that both strains react to each other, encounter enhanced competition for phosphorus and iron and shift towards alternative metabolic pathways.

**Sphingomonas Leaf257, a plant-derived xylan degrader and sugar generalist**. Our proteomics data suggested that *Sphingomonas* Leaf257 explores new niches when experiencing species competition. In particular, the data for co-colonization compared to mono-colonization indicated an increased utilization of xylan from the cell wall (Fig. 3a and Supplementary Data 4), a polysaccharide predominantly made up of xylose. To verify the ability of *Sphingomonas* Leaf257 to metabolize plant-derived xylan, we performed growth assays in a minimal medium containing xylan as the sole carbon and energy source. Indeed, we observed growth both in liquid culture as well as on plates, confirming the ability of *Sphingomonas* Leaf257 to degrade xylan (Fig. 3b). Growth on xylan requires the secretion of xylan-degrading enzymes. To test for the presence of such proteins, we sampled the spent medium from liquid cultures with xylan, concentrated it and analyzed it using SDS-PAGE. We observed prominent bands at around 55 kDa upon growth on xylan, but not when glucose was the carbon source (Fig. 3c). LC-MS analysis resulted in the identification of several proteins, six of which were enzymes involved in the degradation of larger polysaccharides such as xylan and its subsequent metabolization (Fig. 3d). One of these enzymes was already identified as significantly induced during co-colonization

(ASF14_10230) (Fig. 3a) and three others were detected *in planta* (ASF14_10220, ASF14_10225, ASF14_10250) (Fig. 3d). The three xylan-degrading enzymes, ASF14_10230, 10250, and 10340, contained a signal peptide, indicating that they are indeed secreted to degrade the xylan polysaccharide. In contrast, both the xylose isomerase and xylulose kinase as well as the glycoside hydrolase did not contain a predicted signal peptide. The presence of these enzymes in culture supernatants might thus be due to partial cell lysis and accumulation in the medium. Nonetheless, their abundance in a minimal medium with xylan is further suggestive of the metabolization of xylose derived from xylan through the pentose phosphate pathway.

The degradation of xylan results in the release of mostly xylose but also additional carbohydrates, including C6 sugars that form side chains on the β-1,4-xylose backbone[77], as mentioned above. To test if *Sphingomonas* Leaf257 prefers certain sugars, we determined the growth on different substrates and their combinations in liquid culture. In addition, we took samples of the supernatant for HPLC analysis. We found that *Sphingomonas* Leaf257 is able to grow on xylose, glucose, and galactose and co-consumes these sugars (Supplementary Fig. 5). We did not observe evidence for the accumulation of incompletely metabolized intermediates such as pyruvate, acetate, or other byproducts during growth on the different sugar substrates. Our data therefore indicate that *Sphingomonas* Leaf257 is a sugar generalist able to consume the carbohydrates derived from xylan degradation.

***Rhizobium* Leaf68 requires PhoB for phyllosphere fitness and is facilitated by *Sphingomonas* Leaf257 to replete its biotin pool**. Next, we enquired fitness traits of *Rhizobium* Leaf68 and the molecular basis for its enhanced growth upon co-colonization by *Sphingomonas* Leaf257. *Rhizobium* Leaf68 (unlike *Sphingomonas* Leaf257) was amenable to genetic manipulation and we selected five genes for mutagenesis (Fig. 4a). We chose the gene encoding the PhoB transcriptional regulator (ASF03_21155) and the gene encoding the GlpR repressor (ASF03_11310) to test whether the underlying regulons are important for phyllosphere colonization in general or play a role in the interaction with *Sphingomonas* Leaf257. We also mutated *adh* encoding for a predicted alcohol dehydrogenase (ASF03_19755). To investigate the relevance of the glyoxylate cycle *in planta* we generated a mutant in *aceA* encoding isocitrate lyase (ASF03_04985). In addition, we tested BioB (ASF03_03440), an enzyme involved in biotin biosynthesis that was induced *in planta* in mono-colonization but reduced in presence of *Sphingomonas* Leaf257. With the exception of the *phoB* mutant the generated *Rhizobium* Leaf68 mutants grew similarly compared to the wild type during growth on minimal medium with glucose (Supplementary Fig. 6). As expected, the isocitrate lyase mutant was unable to grow on acetate (Supplementary Fig. 6g) confirming the essentiality of *aceA* for the growth on the substrate[78,79]. Next, we tested whether each mutant retained the ability to colonize the plant and achieved comparable population sizes compared to *Rhizobium* Leaf68 wild-type, which was the case (Fig. 4b) (Kurskal–Walllis, *P*-value = ns). Although cell numbers after plant colonization were comparable, strains might still be at a growth disadvantage when competing with the parental strain. To test this, we conducted *in planta* assays in which we competed the mutants against the wild-type strain using an initial 1:1 ratio in the inoculum. While the isocitrate lyase (ASF03_04985) mutant showed no apparent fitness defect, the alcohol dehydrogenase and GlpR mutants showed a small competitive disadvantage when challenged against the wild type (Fig. 4c). The *bioB* and *phoB* mutants showed a strong decrease in competitiveness against the wild type (Fig. 4c). Thus, our data underscore that BioB and PhoB, which are known to be involved in environmental adaptation of Salmonella[80,81], are required for fitness during phyllosphere colonization of *Rhizobium* Leaf68.

Biotin synthase (BioB) catalyzes the last step in biotin biosynthesis. To determine whether the strain has the potential to produce biotin, we searched the genome of *Rhizobium* Leaf68 using BLAST for additional genes encoding enzymes involved in the biosynthesis of the coenzyme. We found that the *bioD* gene encoding ATP-dependent dethiobiotin synthase was absent, suggesting that the strain is an auxotroph for biotin - or the precursor dethiobiotin. We confirmed that *Rhizobium* Leaf68 was unable to grow on minimal medium containing glucose without vitamin addition (Supplementary Fig. 6a–c). We observed that Leaf68 wild-type was able to grow with biotin and dethiobiotin alike, and that the *bioB* mutant required biotin for growth (Supplementary Fig. 6d, e). We then examined whether the *bioB* mutant can be rescued in competition against the wild type *in planta* upon external biotin supplementation compared to mock treatment. Indeed, the supplementation with biotin rescued the *bioB* mutant during competition with the wild type strain (Fig. 5a), further confirming the relevance of this trait during plant colonization.

The findings described above suggested that dethiobiotin is available *in planta* and may induce *bioB* expression to catalyze the last step in biotin biosynthesis. BioB is an instable enzyme that is partially consumed during the reaction[82,83]. It was also shown that upon addition of dethiobiotin a significant fraction of the protein is degraded[84]. Therefore, the reduction in BioB in *Rhizobium* Leaf68 upon co-colonization with the *Sphingomonas*

Leaf257 (Supplementary Data 5; Fig. 4a) may reflect a higher degradation of BioB due to an increase in the dethiobiotin pool available to *Rhizobium* Leaf68. We thus set out to investigate whether the *bioB* mutant was impaired in its interaction with *Sphingomonas* Leaf257. Specifically, we sought to test whether the *Rhizobium* Leaf68 *bioB* mutant mitigates the negative interaction towards *Sphingomonas* Leaf257 and the failure of the latter to benefit the *Rhizobium* strain. Remarkably, we observed a significant loss of the interaction with the *Rhizobium* Leaf68 *bioB* mutant (Fig. 5b), indicating that an increased dethiobiotin pool provided by *Sphingomonas* Leaf257 is responsible for the significant increase in the *Rhizobium* Leaf68 population during co-colonization (Fig. 2b). In addition, and to complete the analysis of the *Rhizobium* Leaf68 with respect to its interaction with *Sphingomonas* Leaf257, we also tested the *Rhizobium* Leaf68 *glpR*, *phoB*, *aceA* and *adh* mutants. For the *glpR* mutant, we observed a significant reduction in population size during co-colonization with *Sphingomonas* Leaf257 (Supplementary Fig. 7a). This suggests that GlpR is relevant for the fitness of *Rhizobium* Leaf68 in competition with the wild type or *Sphingomonas* Leaf257 *in planta* (Fig. 4c, Supplementary Fig. 7a). In contrast, the *Rhizobium* Leaf68 *phoB* mutant showed no effect on the interaction between *Rhizobium* Leaf68 and *Sphingomonas* Leaf257, suggesting that PhoB is relevant for fitness *in planta* compared to the wild type but not for the biotic interaction between Leaf68 and Leaf257 (Fig. 4c, Supplementary Fig. 7b). Neither of the other two *Rhizobium* mutants, *aceA* and *adh*, showed an effect when tested in competition (Supplementary Fig. 7c, d).

After establishing the dethiobiotin dependence in the interaction of *Rhizobium* Leaf68 with *Sphingomonas* Leaf257, we tested whether *Sphingomonas* Leaf257 was able to complement *Rhizobium* Leaf68 in vitro in minimal medium (Supplementary Fig. 8c, d). Indeed, the prototroph *Sphingomonas* Leaf257 was able to sustain growth of *Rhizobium* Leaf68 wild type in vitro in the absence of biotin supplementation (Supplementary Fig. 8c, d). Finally, we tested whether exogenous application of biotin to the phyllosphere was sufficient to recover the interaction of the *Rhizobium* Leaf68 *bioB* mutant and *Sphingomonas* Leaf257. Notably, exogenous application of biotin restored the phenotype of the *Rhizobium* mutant interacting with *Sphingomonas* Leaf257 to wild-type levels (Fig. 5c). In the control treatment (mock) the interaction between *Rhizobium* Leaf68 wild-type and *Sphingomonas* Leaf257 remained unchanged (Fig. 5b). The addition of biotin and its precursor dethiobiotin significantly increased the abundance of *Rhizobium* Leaf68 wild-type during mono-colonization of the plant (Supplementary Fig. 8a), while *Sphingomonas* Leaf257 was unaffected by the treatments (Supplementary Fig. 8b). As expected, the Leaf68 *bioB* mutant increased its abundance significantly upon the addition of biotin but remained unchanged during dethiobiotin addition (Supplementary Fig. 8a).

Taken together, our results indicate that in the presence of *Sphingomonas* Leaf257, the available dethiobiotin pool on the leaf surface is increased such that it enables *Rhizobium* Leaf68 to reach a higher colonization capacity. They highlight that the two-sided interaction of *Rhizobium* Leaf68 and *Sphingomonas* Leaf257 is triggered by vitamin supplementation *in planta* and manifests in dynamic niche displacement of the latter toward exploiting an additional substrate, namely xylan as revealed by protein allocation (Fig. 3).

## Discussion

Gnotobiotic model systems provide the opportunity to address fundamental ecological questions and dissect the mechanism of

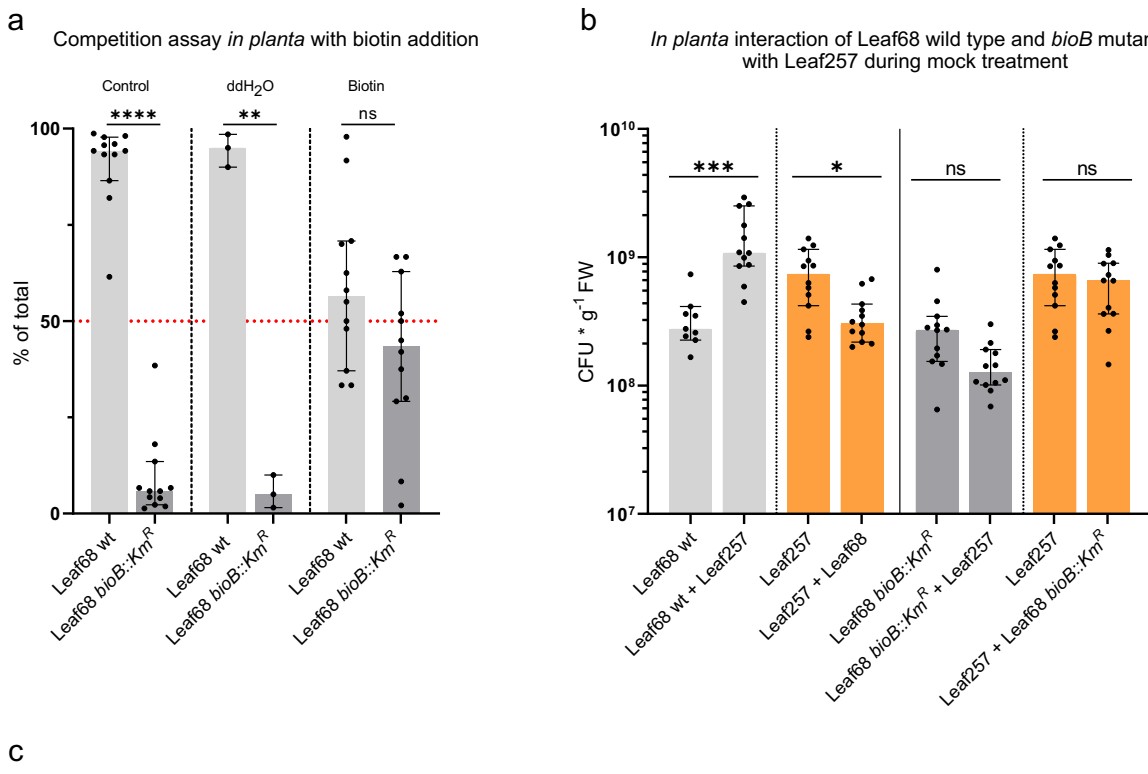

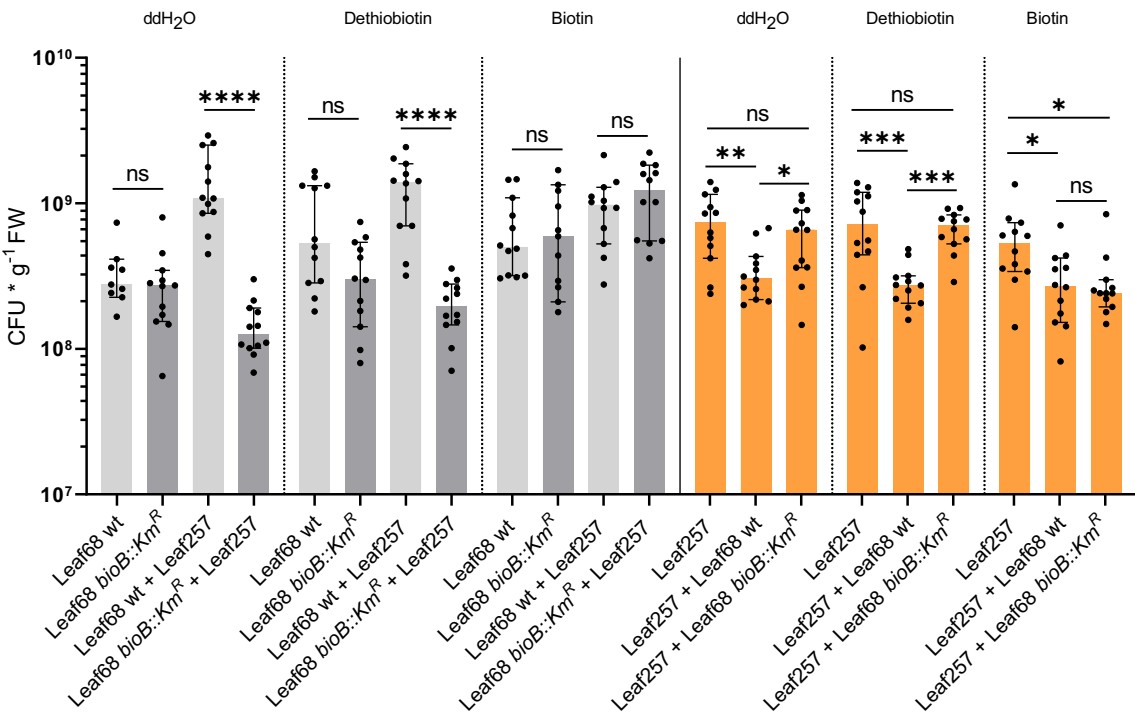

**Fig. 5 *In planta* biotin complementation assays and interaction with *Sphingomonas* Leaf257. a** Competition assay depicting the percentage (%) of total population of *Rhizobium* Leaf68 wild type (light gray) versus Leaf68 *bioB::Km^R* (dark gray) without treatment (control), mock treatment (ddH₂O) and biotin treatment post inoculation (p.i) (see methods). For statistical analysis a two-sided paired t-test was performed. A 1:1 mixture is indicated at 50% with the red dotted line. **b** Depiction of the bacterial colonization (CFUs per gram fresh weight) for Leaf68 wild type (light gray), Leaf68 *bioB::Km^R* (dark gray), *Sphingomonas* Leaf257 (orange) during mono- and co-colonization and mock (ddH₂O) treatment of the plant. **c** Bacterial colonization (CFUs per gram fresh weight) of Leaf68 (light gray), Leaf68 *bioB::Km^R* (dark gray) and Leaf257 (orange) during mono- and co-colonization treatment with either ddH₂O, dethiobiotin or biotin *in planta* (see "Methods"). Each data point represents the CFUs of a single plant with a total of 12 plants for this experiment, except for the mock (ddH₂O) in A due to a contamination. In all graphs, the median and 95% confidence interval are shown. For (**b**, **c**), statistical analysis was performed using the Kruskal–Wallis test (*P*-values: *$p$-value < 0.05, **$p$-value < 0.01, ***$p$-value < 0.001, and ****$p$-value < 0.0001). Source data are provided as a Source data file.

bacterial interactions in native-like conditions in a controlled environment[27,28]. Omics approaches such as transcriptomics, proteomics and metabolomics offer the possibility to record specific phenotypic states at given time points[25,44,47,49,85,86]. Here, we characterized the in situ phenotype and interaction of two genome-sequenced bacterial isolates of prevalent bacterial taxa from the *At*-LSPHERE[29] by mass spectrometry-based proteomics and revealed their ecology and adaptive strategies underlying their coexistence in the phyllosphere.

We first investigated both bacteria colonizing the plant in mono-colonization to establish their general ecological niche occupancy. The proteomics data of the bacteria, *Rhizobium* Leaf68 and *Sphingomonas* Leaf257, *in planta* compared to artificial media as reference indicated adaptive traits that were in line with previous observations in the phyllosphere[25,42,44,47,49,67]. We observed the expression of species-specific but also shared adaptive strategies towards the phyllosphere environment (Supplementary Fig. 4). Overlapping ecological niches makes the pair interesting to study their coexistence with regard to niche separation to evade competition, here referred to as dynamic character displacement. Indeed, both bacteria reacted to each other upon co-colonization, observable in changes in absolute cell abundance and differential expression in the proteome (Fig. 2). Niche separation can be observed in *Sphingomonas* Leaf257, which allocates its proteome towards the degradation of plant-derived xylan and simultaneous utilization of the derived monosaccharides. The niche versatility of *Sphingomonas* Leaf257 is further underscored by the induction of numerous TonB-dependent receptors and efflux systems also observed in other *Sphingomonas* under competitive environmental conditions[25]. The tendency of *Sphingomonadaceae* to utilize a wide plant-associated substrate range might be an evolutionary result of their consistent phyllosphere colonization across different plant species[20,26,87,88]. Especially, the ability to degrade plant-derived xylan revealed in this study (Fig. 3, Supplementary Fig. 5) might be a common resilience trait, enabling *Sphingomonas* to explore new niches during co-colonization with other phyllosphere microbiota members. While the accessibility of xylan to leaf epiphytes is not fully clear, the leaf contains areas where the cuticle thickness can vary greatly[89], and accessibility might be higher at veins, leaf margins and the basis of trichomes.

During co-colonization, *Rhizobium* Leaf68 induces the glyoxylate cycle and allocates proteome resources towards the beta-oxidation of fatty acids. Notably, fatty acids are important structural components and precursors of the plant cuticle making it a potentially resource for nutrients. Overall, the proteomes of both species suggest that both bacteria undergo a niche separation to evade competition. Since changes were inducible and occurred specifically upon coexistence of the two species we refer to dynamic character displacement rather to extend the concept of ecological character displacement that refers to the genetic difference of strains[2].

We found that facilitation by vitamin sharing contributes to the increase of the *Rhizobium* Leaf68 cell numbers in the presence of *Sphingomonas* Leaf257 (Fig. 2b). We identified the biotin precursor dethiobiotin to be crucial for this population gain: In contrast to wild-type cells, the *Rhizobium* Leaf68 *bioB* mutant did not profit from the presence of *Sphingomonas* Leaf257 nor from dethiobiotin addition but showed an increase in cell numbers after addition of biotin to the phyllosphere (Fig. 5c, Supplementary Fig. 8a). To understand the nutritional and molecular basis facilitating microbe-microbe interactions, further studies are needed to elucidate the metabolic profile of the phyllosphere regarding free vitamin pool sizes on leaves. However, the finding that the addition of biotin or its precursor dethiobiotin to plants increases the population of *Rhizobium* Leaf68 indicates that the

vitamin is growth limiting, but that some biotin or precursor is available. The microbiota, or as tested here, *Sphingomonas* Leaf257 helps to supply the biotin precursor dethiobiotin rather than biotin as revealed by the importance of biotin synthase BioB in the interaction and fitness gain of *Rhizobium* Leaf68. Passive leakage might occur for dethiobiotin as observed for biotin[90]. It is currently unclear why dethiobiotin rather than biotin becomes available. An explanation might be the chemical stability of dethiobiotin compared to biotin. Under mild conditions and in oxygenic environments, such as the phyllosphere, biotin is readily oxidized to biotin sulfoxide, while dethiobiotin is not, due to the absence of sulfur. In line with this conclusion is the observation that *Rhizobium* Leaf68 induced a biotin sulfoxide reductase during mono-colonization but did not change its expression during co-colonization with Leaf257 (Supplementary Data 2 and 5). This suggests that *Rhizobium* Leaf68 scavenges biotin sulfoxide that could become available in the phyllosphere but relies on the dethiobiotin obtained from *Sphingomonas* Leaf257.

Metabolic dependencies are thought to strengthen the interaction among bacteria by sharing metabolites, leading to adaptive gene loss[91–96]. Although vitamin sharing by *Sphingomonas* Leaf257 was shown to be sufficient to explain the growth facilitation of *Rhizobium* Leaf68, also the release of extracellular enzymes for xylan degradation by *Sphingomonas* Leaf257 could be used as a shared public good helping other species through active niche construction, making *Sphingomonas* Leaf257 a potential key stone species. In a previous study *Rhizobium* Leaf68 was not impacted by the presence or absence of one specific strain in a 62 member community[27]. However, many strains might share vitamins, making the strain the most commonly detected one in a higher complexity microbiota. While *Sphingomonas* Leaf257 was not tested in this particular complex community, another *Sphingomonas* strain, Leaf231, was identified as a key-stone strain.

In summary, we used MS-based proteomics to characterize metabolic shifts and identified the phenotypic plasticity of interacting species, here referred to as dynamic character displacement. While our data demonstrate that both bacteria respond to each other and become more dissimilar in their respective proteome constitution under mono- versus co-colonization, investigations on the microscale level will remain to be conducted. Specifically, the heterogeneous nature of leaves[22,41,97–99] will give rise to subpopulations with each potentially expressing different adaptations to their local microenvironment. While our bulk approach captured changes upon co-colonization, further investigation at the single-cell level will be of interest. Here, microscopy-based approaches such as FISH to visualize bacterial populations in situ[40,41,100] and fluorescent reporters to monitor expression with spatial resolution[22,41,97] or par-seqFISH[101] could be applied. Our ability to combine in situ phenotyping and relate it to the genetic potential of phyllosphere colonizing bacteria leads us to speculate that the observed dynamic character displacement, as described here, is an important mechanism during the formation of stable microbial communities, and supports the notion that both genetic diversity and phenotypic diversity promotes coexistence, which remains underexplored[10,17,18]. This study contributes to our understanding about how bacteria coexist in complex, oligotrophic environments. Such understanding is a prerequisite for the design of stable, biologically relevant synthetic communities. Due to redundant functions of microbiota members, a bottom-up approach of studying pairs of interactions under controlled conditions represents a way to uncover bacteria-bacteria interactions and together with higher complexity synthetic communities and environmental studies will contribute to a better understand the dynamics and structure of

biological diverse communities, their assembly and ultimately their astonishing persistence.

## Methods

**Strains and growth conditions**. The two bacterial strains *Sphingomonas* Leaf257 and *Rhizobium* Leaf68 were selected from the *At*-LSPHERE collection[29] based on their colonization capacity and previous data collected[27,55]. Both strains were grown on R-2A-agar (Sigma-Aldrich, Buchs, Switzerland) supplemented with 0.5% v/v methanol (R2A + M) for 3 days at 22 °C prior usage.

**Mutant strain construction in *Rhizobium* Leaf68**. The list of primers used is provided in Supplementary Data 6. For the generation of deletion mutants of Leaf68 the flanking regions of ~650 bp were amplified using the upstream (HR1) and downstream (HR2) primers for each mutant respectively (Supplementary Data 6) followed by digestion with MunI/KpnI (upstream) and KpnI/NsiI (downstream) and inserted into the pREDSIX vector[102]. The construct was cleaved with KpnI between the two flanking regions of the respective gene and a kanamycin-resistances cassette (KmR) was inserted that has been cut from pRGD-KmR[102] with KpnI. The orientation of the cassette was confirmed by PCR with primers binding the flanking region (OR/OF, this study) and kanamycin cassette (Kan-2/-4 described previously by Ledermann et al., 2016[102]). Both possible insertion directions were selected, and electroporated into *Rhizobium* Leaf68. For this, 100 µl competent cells were pulsed with 0.5 µg plasmid DNA at 2.2 kV in a 1-mm-gap cuvette (MicroPulser BIO-RAD). After 5 h of recovery in half LB medium at 28 °C cells were plated onto R2A agar plates containing 50 µg/mL Kanamycin. The insertion deletion mutants of all clones were confirmed by PCR using primers inside (IF/IR) and outside (OF/OR) the flanking region, respectively.

**Growth assay in micro-titer plates**. Overnight pre-cultures were grown in 20 mL minimal medium with 20 mM glucose (MM + G) in 100 mL baffled shake flasks at 28 °C. For *Rhizobium* Leaf68 50 µM of biotin, pantothenic acid and niacin were added to complement the auxotrophies. For the validation of the auxotrophies (Supplementary Fig. 5a-e) another pre-culture in MM + G without vitamins was inoculated. After 2-3 doublings cells were collected. To remove the remaining substrate, cells were spun down at 3220 × g at room-temperature for 10 min. The pellets were washed with two volumes equivalents of 10 mM MgCl₂ and re-suspended in 10 mM MgCl₂. Next, the OD₆₀₀ was adjusted to 0.5 to inoculate the main cultures in 96-well pates (ThermoFischer Scientific Nunclon 96 Flat Bottom Transparent Polystyrol). The plates contained a minimal medium (180 µL) containing the respective carbon source and the three vitamins (biotin, pantothenic acid, niacin) together, in different binary combinations, or each alone. The 96-well plates were inoculated with 20 µL of culture to reach a final OD₆₀₀ of 0.05. The OD₆₀₀ was measured every 10 min using a Tecan Infinity M200 Pro spectro-photometer (Tecan) with a bandwidth of 9 nm and 25 flashes. The plates were shaken orbital with 1 mm amplitude for 15 s between measurements while incubating at 28 °C. Growth curves were analyzed using the Python-based tools pandas version 1.0.1 (https://pandas.pydata.org/) and matplotlib version 3.1.3 (https://matplotlib.org/stable/index.html).

**Gnotobiotic growth conditions of *Arabidopsis thaliana* plants**. *Arabidopsis thaliana* Columbia (Col-0), *bak1-5/bkk1-1*[103], *jar1-1*[104], *rbohd*[105], and *npr1-1*[106] seeds were surface sterilized according to Schlesier et al.[107] and stratified at 4 °C in the dark for 4 days. Sterile seeds were sown in 24 well cell-culture plates (TPP Techno Plastic Products AG, Switzerland) on full strength Murashige and Skoog (MS) medium (pH5.8, including vitamins) (Sigma-Aldrich, Buchs, Switzerland) supplemented with plant agar (Duchefa, Haarlem, Netherlands) and 3% w/v sucrose. Plants were grown for 1 week under long-day growth conditions (16-h-photoperiod) before switching to short-day conditions (9-h photoperiod) for the rest of the experiment. The temperature in the chamber (CU-41L4, Percival Perry, USA) was set to 24 °C during light and 22 °C during dark cycles at a constant relative humidity of 65% rh.

**Plant inoculation**. Three days prior to inoculation, the bacterial strains or mutants were pre-grown on R2A + M agar plates. For inoculation after 10 days of plant growth a sterile 1 µl plastic loop was used to collect cell material from each strain and cells were re-suspended in 1 mL of 10 mM MgCl₂. The tubes containing the bacteria were vortexed and optical density (OD₆₀₀) was adjusted to a final concentration of 0.02 per strain. Notably, the mixture of both had a final OD₆₀₀ of 0.04 with both strains having a concentration of 0.02. The gnotobiotic 10-day-old plants were inoculated with 10 µL of bacterial solution by pipetting. At this growth stage, the plant has four leaves. Each leaf was treated with a drop containing 2 µL, an additional drop of 2 µL was distributed in the middle of the plant. Axenic control plants were mock-inoculated with 10 µL of 10 mM MgCl₂. Plants were further grown for 11 days (total plant age at harvest time point: 21 days).

**Chemical complementation with biotin *in planta***. Prior the plant treatment a biotin stock of 1 mM in ddH₂O was prepared, sterile filtrated and further diluted to 10 µM in ddH₂O. Plants were treated at two time points with the first 30 min after

inoculation and the second 3 days later. For treatment either 10 µL 10 µM biotin (treatment) or 10 µL ddH₂O (mock) or 10 mM MgCl₂ (mock) were carefully pipetted to the phyllosphere without touching the leaves.

**Microscopy**. Bacterial strains *Rhizobium* Leaf68 and *Sphingomonas* Leaf257 expressing a fluorescent protein (mCherry) (Lab stock) were grown on R2A + M and prepared for plant inoculation as described above. Prior to inoculation, strain Leaf68 was fluorescently labeled using Alexa Fluor™ 488 NHS Ester (Thermo-Fischer Scientific) following the manufacturer's instructions. The strains were adjusted to a final OD₆₀₀ of 1 (Supplementary Fig. 1a) or 0.1 (Supplementary Fig. 1b-d) and mixed prior to inoculation. Inoculated plants were grown for 24, 48, and 72 hours in the plant growth chamber at the short-day settings described above. Single leaves were detached and directly mounted on a standard cover slip. Images were acquired using a standard fluorescence microscopy setup (Zeiss Axio Observer Z1, Hamamatsu Orca-ER) using a ×40 objective. Fiji software[108] (ImageJ, v.2.0.0) was used for linear intensity adjustments and to insert scale bars.

**Enumeration of bacterial growth in the phyllosphere**. To determine bacterial colonization after 11 days post-inoculation colony-forming units (CFUs) were measured using a leaf-wash protocol according to Vogel et al.[51]. For the leaf washing, the above-ground parts were separated from the roots using a sterile scalpel. The resulting phyllosphere of single plants was transferred into phosphate buffer (pH 7, 100 mM) containing surfactant (Silwet L-77, 0.2% v/v) (Leu + Gygax AG, Birmensdorf, Switzerland). Bacteria were washed off as described previously[51]. Briefly, tubes were shaken for 2 × 7.5 min at 25 Hz in a Qiagen Tissue Lyser II (Qiagen) and sonicated for 5 min in an ultrasonic cleaning bath (Branson Ultra-sonics). The leaf wash was vortexed and plated on R2A + M -agar using a tenfold dilution series (10⁰-10⁻⁸). After 2-3 days of incubation at 22 °C CFUs per g plant fresh weight were determined. On plates, both strains could be distinguished by eye according to the color of the colonies, with *Sphingomonas* Leaf257 forming yellow and *Rhizobium* Leaf68 forming white colonies. Bacterial abundance data were analyzed using Prism 9.2 (GraphPad).

**Competition Leaf68 wild type and mutants**. To determine the percentage of total cells we inoculated and enumerated bacterial growth as described above and plated the leaf was on R2A + M to recover the total CFUs and R2A + M containing 50 µg/mL Kanamycin to recover the mutant CFUs. For each comparison, we inferred the Leaf68 wild type CFUs by subtracting the total CFUs from the mutant CFUs and plotted the percentage of total for each wild type and mutant.

**Harvest of phyllosphere bacteria for MS-based proteomics**. To recover bacteria from the phyllosphere of *A. thaliana*, the phyllosphere of 10 plants was separated from the rhizosphere using flame sterilized scalpels and forceps and transferred into 50 mL Falcon tubes containing 25 mL ice-cold TE-P buffer (10 mM Tris-HCl, 1 mM EDTA, pH 7.5). The epiphytic bacteria were washed off leaves using a previously established protocol that minimizes plant contamination[44]. Briefly, it consists of alternating cycles of vortexing and sonication as well as a soft filtration step through a nylon mesh to remove large plant particles, followed by the concentration of the bacterial fraction. For each biological replicate the enriched cell pellet corresponding to 20 plants was pooled, snap frozen and stored at −80 °C until further processing.

**Proteomics of bacteria recovered from the phyllosphere**. In total, four biological independent replicates were generated for MS-based proteomics analysis. As a reference to the *in planta* conditions, both *Rhizobium* Leaf68 and *Sphingomonas* Leaf257 were each grown on R2A + M-agar (see above) in the plant growth chamber at the settings described above. Bacterial cell pellets were dissolved and lysed using a lysis buffer containing 100 mM ammonium bicarbonate, 8 M urea and 1× cOmplete EDTA-free protease inhibitor cocktail (Sigma-Aldrich, Buchs, Switzerland) and indirect sonication (3 × 1 min, 100% amplitude, 0.8 cycle time) in a VialTweeter (HIFU, Hielscher, Teltow, Germany)[45,109]. Insoluble parts were removed by centrifugation at 13,000 g for 15 min at 4 °C. Protein concentration of supernatant was determined using the Pierce BCA assay kit (Thermo Fischer Scientific, Reinach, Switzerland) according to the manufacturer's instructions. Protein disulfide bonds were reduced and cysteine residues were alkylated as described previously[45] using 5 mM tris(2-carboxylethyl)phosphine (TCEP, Sigma-Aldrich, Buchs, Switzerland) and 10 mM iodoacetamide respectively (IAA, Sigma-Aldrich, Buchs, Switzerland). Prior to protein digestion, samples were diluted 1 to 5 with freshly prepared 50 mM ammonium bicarbonate buffer to reduce the urea concentration below 2 M. Sequencing grade modified trypsin (Promega AG, Dübendorf, Switzerland) was added at a trypsin to protein ratio of 1:50 and protein digestion was carried out overnight at 37 °C with shaking at 300 rpm. After incubation, trypsin was inactivated using heat incubation in a tabletop shaker at 95 °C for 5 min followed by acidification through addition of formic acid to an approximate final concentration of 1%. Insoluble parts were removed by centrifugation at 20,000 g for 10 min and the supernatant was desalted using Sep-Pak Vac C18 reversed phase columns (Waters Corporation, Baden-Dättswil, Switzerland) as previously described[109] and dried under vacuum. The samples were

re-solubilized in 3% acetonitrile (ACN) and 0.1% formic acid (FA) to a final concentration of 0.1-1.0 mg mL$^{-1}$.

**MS analysis.** Mass spectrometry analyses were performed on an Orbitrap Fusion Tribrid mass spectrometer (Thermo Fischer Scientific) equipped with a digital PicoView source (New Objective, Littleton, USA) and coupled to an M-Class ultraperformance liquid chromatography (UPLC) system using an ACN/water solvent system containing two channels with 0.1% FA (v/v) in water for channel A and 0.1% FA (v/v), 99.9% ACN (v/v) for channel B. For chromatographic separation 2 μL peptide sample (at a concentration of 0.5 μg μL$^{-1}$) was loaded on a nanoEase M/Z Symmetry C18 trap column (100 A, 5 μm; 180 μm × 20 mm, Waters) followed by a nanoEase M/Z HSS T3 column (100 A, 1.8 μm; 75 μm × 250 mm, Waters). Peptide samples were separated at a flow rate of 300 nL min$^{-1}$ using the following gradient: 2% to 5% B in 2 min, 25% B in 93 min, 35% B in 10 min, and 95% B in 10 min. The MS was operated in data-dependent acquisition and top-speed mode with a maximum cycle time set to 3 s. Full-scan MS spectra were acquired in the Orbitrap analyzer with a mass range of 300-1500 $m/z$ and a resolution of 120k with an automated gain control (AGC) target value of $5 \times 10^5$. Precursor ions were isolated with a window of 1.6 $m/z$ and fragmented using higher energy collisional dissociation (HCD) with a normalized collision energy of 30%. For fragmentation only precursor ions with charge states from +2 to +7 and a signal intensity of at least $5 \times 10^3$. Fragment ion spectra (MS/MS) were acquired in the Ion trap operated in rapid scan mode with an AGC value of 8000 using a maximum injection time of 300 ms. The dynamic exclusion was set to 25 s. Sample measurements were acquired using internal lock mass calibration on $m/z$ 371.10124 and 445.12003.

**Database search and quantitative analysis.** Label-free precursor (MS1) intensity-based quantification was performed using Progenesis QI for proteomics (Nonlinear Dynamics, v.4.0) as previously described[45,110]. Briefly, for all samples the automatic alignment was reviewed and manually adjusted before normalization. From each Progenesis peptide ion (default sensitivity in peak picking) a maximum of the top five tandem mass spectra were exported as mascot generic file (*.mgf) using the charge deconvolution and deisotoping option and a maximum number of 200 peaks per MS/MS. The mascot generic files were searched using the Mascot Server (Matrix Science, v.2.6.2) against a decoyed and reversed protein sequence database containing the 4341 annotated proteins of *Rhizobium* Leaf68 (NCBI Taxon ID: 1736231) and the 4024 annotated proteins of *Sphingomonas* Leaf257 (NCBI Taxon ID: 1736309) concatenated with the Arabidopsis Information Resource (TAIR) database (release TAIR10) and 260 known mass spectrometry contaminants. Parameters for precursor ion tolerance and fragment ion tolerance were set to ± 10 ppm and ± 0.5 Da, respectively. The search parameters were as followed: trypsin digestion (two missed cleavages allowed), as fixed modification of the carbamidomethylation of cysteine and as variable modification of the oxidation of methionine, carbamylation of the N-terminus and lysine. The mascot search was imported into Scaffold (Proteome Software, v.4.8.9) using 5% peptide and 10% protein false discovery rate (FDR) as thresholds. The scaffold spectrum reports were imported into Progenesis QI. Normalization was performed on all precursor ions of the corresponding strain. Notably, in the mixture and artificial mixture conditions normalization was performed on all precursor ions from Leaf68 or Leaf257 respectively, i.e. to quantify the Leaf68 proteome in the mixture we only considered precursor ions specific for proteins from Leaf68 for normalization and vice versa. For protein quantification, the three most abundant peptide ions (Hi-3 approach) were used for protein quantification. Only proteins with two or more unique peptides detected were considered for quantification. For statistical testing, one-way ANOVA was applied and the resulting $P$-values were corrected using the Benjamini-Hochberg procedure directly in Progenesis QI resulting in q-values. If not indicated otherwise the general cutoffs for significantly regulated proteins were q-value < 0.05 and Log$_2$ fold-change (FC) >1.5 or < -1.5.

**Comparison of orthologues protein groups in Leaf257 and Leaf68.** The annotated genomes for both strains were obtained from the PATRIC database[111]. Subsequently, the genomes were annotated with orthologous groups (OGs) using eggNog v.4.5[112]. In order to identify shared OGs, the proteomics data of both bacteria colonizing the plant alone during mono-association was compared by identifying shared regulated OGs. For the OG comparison only proteins/OGs detected with at least 3 unique peptides and a q-value < 0.05 and Log$_2$ fold-change >1.5 or < −1.5 in both bacteria were considered.

**Experimental investigation of sugar preferences in *Sphingomonas* Leaf257 using HPLC.** The sugar utilization characterization of Leaf257 was performed in a volume of 20 mL in 100 mL baffled shake flasks at 22 °C and 160 rpm. in a Minitron Incubator (Infors HT). The minimal media contained either of the three sugars (10 mM), glucose, xylose and galactose (Sigma-Aldrich, Buchs, Switzerland) alone as sole carbon source or binary combinations of each. To estimate bacterial growth OD$_{600}$ of all conditions was measured over eight-time points and simultaneously 1 mL of the culture was used for further analysis of the supernatant. Supernatant analysis for each sample was performed by high-performance liquid chromatography using an Ultimate 3000 UHPLC device (Thermo Fischer

Scientific) equipped with a Rezex ROA Organic Acid H + column (7.8 × 300 mm; Phenomenex) as analytical column and a RI-detector (RefractorMax521). The mobile phase was 2.5 mM H$_2$SO$_4$ at a flow rate of 0.6 mL min$^{-1}$ and the conditions were isocratic. The sample injection volume was 10 μL and the refractive index (RI) was monitored for metabolite detection.

**Purification and identification of xylan-degrading enzymes in *Sphingomonas* Leaf257.** To test for the secretion of xylan-degrading enzymes, Leaf257 was grown in minimal medium containing 10 mM xylan (Roth AG, Arlesheim, Germany). Supernatant samples were taken from the liquid cultures, centrifuged and sterile filtrated to remove bacterial remains. The supernatant was concentrated in a centrifugal filter with a molecular weight cutoff of 10 kDa (Amicon Ultra, Merck Millipore) and analyzed using SDS-PAGE analysis revealing a protein band specific for growth on minimal medium with xylan (Fig. 3c), which was identified by in-gel digestion and LC-MS (Fig. 3c, d).

**Reporting summary.** Further information on research design is available in the Nature Research Reporting Summary linked to this article.

## Data availability

The data obtained from the MS-based proteomics approach has been deposited to the ProteomeXchange Consortium (http://proteomecentral.proteomexchange.org) via the PRIDE partner repository[113] with the dataset identifier PXD026619 and 10.6019/PXD026619 [https://www.ebi.ac.uk/pride/archive/projects/PXD026619]. A list of all identified protein is available from the PRIDE dataset. The data generated in this work are provided in the Supplementary Information and Source Data files.

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

## Acknowledgements

We thank Paola Picotti, Sebastian Pfeilmeier and Christopher Field for helpful discussions. We thank Bernd Roschitzki and Jonas Grossmann from the Functional Genomics Center Zurich (FGCZ) for support with the LC-MS/MS setup. We thank Tim Keys and Corina Mathew for support with the HPLC setup and Christine Vogel and Raphael Ledermann for input into generating the mutants of *Rhizobium* Leaf68. This work was supported by an ERC Advanced grant (PhyMo, no. 668991), a grant from the Swiss National Science Foundation (310030B_201265) and as part of NCCR Microbiomes, a National Centre of Competence in Research, funded by the Swiss National Science Foundation (no. 51NF40_180575).

## Author contributions

L.H., M.A. and J.A.V. designed the study; L.H. led the experimental work; B.A.M. and M.B.-M. contributed to performing the plant experiments; M.B.-M. helped with the mutant generation in *Rhizobium* Leaf68; B.R. contributed to the characterization of the auxotrophies in *Rhizobium* Leaf68; C.G.G. performed the microscopy analysis; L.H. performed the bioinformatics analysis; L.H. and J.A.V. wrote the manuscript with input from all authors.

## Competing interests

The authors declare no competing interests.
