## [Peer Review File · Nature Communications]

REVIEWER COMMENTS

Reviewer #1 (Remarks to the Author):

This manuscript documents the findings of a study that compares the proteomes of two phyllosphere-fit bacterial strains called Leaf68 (*Rhizobium* species) and Leaf 267 (*Sphingomonas* species) as they colonize the *Arabidopsis* leaf surface under gnotobiotic conditions, either by themselves or in the presence of each other.

The data reveal differences that suggest that 1) for both Leaf68 and Leaf 267, the protein expression profile is different between cells grown in planta (i.e. on leaf surfaces) or in vitro (i.e. on an agar plate), 2) there is overlap between Leaf68 and Leaf267 in the kind of protein functions that both express in planta, and 3) both Leaf68 and Leaf 267 express proteins in each other's presence in planta that they do not express in planta when they colonize the leaf surface by themselves.

The authors frame their study in the context of the phenomenon of 'character displacement', which I had not heard of before. The description of how phenotypic plasticity fits into this (lines 52-57) felt a bit too abstract for me, as did the accompanying illustration (Figure 1). I learned more from looking up some of the referenced papers, so I suggest that the authors redo their description and give some concrete examples from the primary literature instead, like this tadpole study:

<https://pubmed.ncbi.nlm.nih.gov/16602305>. It would also help to include one or more relevant studies from the field of microbiology. This would help me and other readers better understand why this paper is different from other papers that have titles like 'Dual transcriptional profiling of xxx and yyy' and that explore interactions between a host and pathogen (e.g. mice and *Toxoplasma gondii*), a host and a beneficial microbe (i.e. wheat and *Azospirillum brasilense*), or as in this paper, two microbes in each other's presence (like two fungi, two bacteria, or a bacterium and a fungus).

As the authors offered a list of all the proteins that are differentially expressed (e.g. lines 154-171 for Leaf 68), I could not help wondering about the averaging effect of the proteomics approach and how that should guide the authors' interpretation of the data. For example, if proteins A, B and C were all found to be induced three-fold in the data, does that mean that proteins A, B and C were three-fold induced in all bacterial cells (which is the default assumption here)? Or could it be that protein A was six-fold induced in 50% of the cells, protein B was 12-fold induced in another 25% of the cells, and protein C 12-fold induced in the other 25%? The latter explanation would be consistent with the known environmental heterogeneity along the leaf surface, where some cells find themselves in an environment where for example protein A is important for survival but proteins B and C are not. The current version of the manuscript is completely lacking appreciation for this alternative explanation. Is there a way to get to confirm this, for example by RT-PCR on individual bacterial cells or by using promoter fusions to genes A, B and C?

It would be extremely useful for the interpretation of the data if the authors also provided insight into spatial distribution of Leaf68 and Leaf267 prior to or at the time of sampling. Do these two strains 'avoid' each other or do they mostly co-habitate? Do they occupy different or overlapping spaces on the leaf? This would be an experiment much like the one done by Monier and Lindow

(<https://doi.org/10.1128/AEM.71.9.5484-5493.2005>). Seeing that it might be difficult to get both strains to express a fluorescent protein, perhaps a FISH approach, like the one developed in the same lab would be a suitable alternative.

The ways in which Leaf68 and Leaf267 are inoculated onto and recovered again from the Arabidopsis phyllosphere are not disclosed until very late in the manuscript (in the Materials and Methods section). For a proper understanding of the results, it is crucial to volunteer this information much earlier in the manuscript. For example, for the proteomics analysis, bacterial cells are retrieved from the leaves by leaf wash and sonication. There is no leaf maceration involved, and so the only cells that are retrieved from the leaves are those that are growing on the leaf surface, not in the leaf interior. Why then do the authors find induction of proteins involved in the catabolism of xylan, which is a component of the plant cell wall, which should not be accessible to bacteria on the leaf surface? Is there another source of xylan on the leaf surface? Is the expression of these proteins truly a response to the presence of xylan, or is there an alternative explanation (e.g. xylose isomerases can interconvert glucose and sucrose; might that be the reason why this protein is produced)? This needs to be addressed in much more detail. Inoculation is achieved by pipetting 10 μ L onto Arabidopsis leaves (line 507). This description is incomplete (how many leaves were those 10 μ L distributed over, did each leaf receive more than one drop? all this should be disclosed). The authors should also explain that by inoculating the two strains as a mixture, rather than inoculating one after the other, makes it much more likely for all cells of the two strains to interact with one another and co-habitat the same locations on the leaf than when cells were sprayed one after the other (which allows for some escape from each other, I imagine).

The authors do not explain how they differentiated Leaf68 CFUs from Leaf267 CFUs for Figures 2A and 2B.

Other comments:

Line 144: seeing that the agar plates were also incubated in light, explain why there light-stress proteins induced in leaf-grown cells.

Line 212: “>|1.5|” : is this correct? I see what the author try to say here, but |1.5| equals 1.5 so “>|1.5|” becomes “>1.5”, which is incomplete. I suggest to write >1.5 or <-1.5.

Line 339 and throughout: check for typos like this one (When should be We), or line 525: plated should be plated and ‘was’ should be removed.

Reviewer #2 (Remarks to the Author):

In this manuscript, Hemmerle et al. present a very interesting study of the molecular character displacement observed in two phyllosphere bacterial strains during co-colonization of a leaf. How the observed bacterial diversity in plant-associated habitats is maintained and to what extent microbial interactions contribute to this is a fascinating research area. I think this paper is very well presented and makes an important contribution to the field. I have a few points that I think need further clarification.

The authors present compelling evidence that resource utilization and ensuing competition are responsible for the dynamics of the two strains in co-colonization versus mono-colonization. However,

I'm still wondering whether some of these dynamics may be driven indirectly by differential responses of the plant host to one or the other strain. For example, in the recent Maier et al. 2021 paper from the Vorholt lab, inoculation with Leaf257 induces considerably more differentially expressed genes in *Arabidopsis* versus Leaf68. Could this differential response of the host plant also be contributing to the increased abundance of Leaf68 in co-localization? I don't know exactly how one might test this, possibly a co-colonization assay on artificial media as was done for the mono-colonization study. Part of this questioning is coming from the idea that perhaps the dynamic proteome changes observed in these two strains in co-colonization may be fairly agnostic to the particular substrate, so long as the resource profile is such that niche overlap exists. One way the authors might address this is by listing in Supp Table 3A and B which of these proteins are also increased/reduced in the artificial media versus in planta.

Figure 1 and lines 410-411: This is a nice narrative but does it really correspond to the observed results? It almost seems like a non-transitive relationship where Leaf68 is the superior competitor but due to its auxotrophy can never completely outcompete Leaf257. And Leaf257 has a greater breadth of sugar use which might allow it to avoid extinction. Can this be tested on media? Presumably Leaf68 is the superior competitor for readily available sugars and Leaf257 can coexist because it can degrade alternative energy sources? The interpretation provided on lines 351-353 suggests that competition is not occurring until Leaf68, via supplementation with Leaf257 dethiobiotin, reaches a certain population size. At which point, Leaf257 switches energy sources at a cost to population growth? Is this a possible hypothesis which can be tested?

Minor comments

Line 25-27: Perhaps make it clearer under what conditions these traits are found, mono or co-occurrence.

Line 33: Change when to we?

Line 37: or the evolution of increased competitive ability?

Line 271: phosphorous?

Line 440: I don't think the authors demonstrate that dissimilarity increases in co-colonization? Dissimilar in terms of what? In terms of an ordination of the proteins found in each strain under mono versus co-colonization?

Line 517: So aboveground plant parts were removed from roots?

Line 519-522: From this description it sounds like just the epiphytic bacteria were collected. Please state this clearly somewhere in the main text e.g. line 63.

How often do members of these two bacterial families co-occur in aggregates on leaves?

Why would biotin synthase be down regulated in Leaf68 during co-colonization?

Figure 3: In A, it's a bit funny to have error bars extend above 100%. Possibly overlay a strip chart to show individual data points?

Reviewer #3 (Remarks to the Author):

In this paper, Hemmerle et al. use proteomics and growth experiments *in vitro* and *in planta* to investigate the metabolic interaction between two common phyllosphere strains. They show that these two strains alter their gene expression patterns plastically when colonising the plant together, compared to when they are inoculated alone. They dig into a few particular proteins and find a facilitative interaction related to the production of the vitamin biotin. They also show metabolic shifts in one species towards the uptake of xylan when the partner is present. Overall, I find this approach to be very powerful, and is a rare opportunity to look into how two microbes interact in a setting that is very close to the natural scenario. The experiments are mostly very complete (see below) and the paper is very well written.

A few things were not very clear to me when reading the paper, though, which I discuss next. I also list a number of general questions below that would be good to address in the introduction or the discussion sections of the paper, and some minor comments.

My main recommendation for improvement is to better outline what the authors think is happening with the biotin interaction. By rereading the text and looking at supplementary Fig. 6A-C (I think it would be helpful to put the message of these growth curves in the main text), I understood that the authors are proposing the following scenario: Leaf257 makes more dethiobiotin available, which Leaf68 wt can make use of to make more biotin, but the Leaf68 bioB mutant cannot. The wt then “steals” dethiobiotin from Leaf257, presumably because it can take it up faster, leading to a competitive advantage and facilitation compared to when it’s alone. If this is the case, then wouldn’t it make sense to supplement with dethiobiotin rather than biotin in Fig. 5C-D? The authors explain that “The addition of biotin significantly increased the abundance of *Rhizobium* Leaf68 wild type during mono colonization of the plant (Figure 5D) and to similar extent as during co-colonization with *Sphingomonas* Leaf257” (it’s mentioned a second time elsewhere “the finding that the addition of biotin to plants increases the population of *Rhizobium* Leaf68”). However, I don’t really see this in Fig. 5D: if I compare the first and the 5th bar, they look similar and there is no statistical test comparing them. Finally to this point, why does *Sphingomonas* not use the dethiobiotin it produces? Isn’t it surprising that it’s freely available in excess?

In short, I am a bit confused about this part and suggest the authors clarify the text a bit and apply the relevant statistical comparisons. If I understood correctly, then I would also recommend them repeating the experiment with the biotin addition but with dethiobiotin.

My other point is regarding the narrative of the paper that the main result is one of plastic niche separation. I agree that there is a clear shift by Leaf257 to metabolise xylan, but from the introduction, I was expecting to see more evidence for niche separation in the genes that are down-regulated. Is there perhaps more to analyse there? Instead, the paper focuses a lot on the biotin interaction, which seems to be less about niche separation and more about exploitation of the biotin precursor. This is all fine, of course. I am just wondering whether the authors may want to reconsider the narrative or make it more nuanced given the interesting results concerning the biotin interaction.

Finally, some more general questions/comments:

- Do these two isolates come from the same plant? Would the studied interaction have co-evolved?
- Are the agar samples mono- and co-colonization as well?
- Some statistical tests are missing: e.g. Fig. 4B,C. According to the text, in Fig. 5B a statistical comparison is needed between Leaf68bioF and Leave58bioB+Leaf257. Related: it's a bit awkward to talk about "a complete loss of the interaction" l. 350 in a statistical sense

And a few questions from someone who is not an expert in proteomic analyses:

- How do you remove plant proteins from analysis? How do you ensure that plant responds the same to mono- and co-colonization?
- Do you correct for population sizes and changes in these when looking at fold-changes?

Minor comments:

l. 64 typo: import -> important

l. 116-119: labelling of Figures 2A, B not quite right

Fig. 2: what is leaf68mix vs. artmix? Specify in caption

l. 275: increased utilisation of xylan compared to what? Mono-colonisation? Fig. 3A: what is Mix and ArtMix?

l. 339: When -> We

l. 354: in respect -> with respect

Fig. 5C, D: I think it's best to combine these panels into 1, it's a bit strange that the data are repeated in both panels

l. 373ff: I don't see the comparison you mention in Fig. 5D, there are no statistics for mono-colonisation in H₂O versus biotin

l. 316: "All[...] mutants grew similarly compared to the wild type during growth on minimal medium with glucose" but Sup. Fig. 6 shows that the phoB mutant grows quite a bit worse

REVIEWER COMMENTS

Reviewer #1 (Remarks to the Author):

This manuscript documents the findings of a study that compares the proteomes of two phyllosphere-fit bacterial strains called Leaf68 (*Rhizobium* species) and Leaf 257 (*Sphingomonas* species) as they colonize the *Arabidopsis* leaf surface under gnotobiotic conditions, either by themselves or in the presence of each other.

The data reveal differences that suggest that 1) for both Leaf68 and Leaf 257, the protein expression profile is different between cells grown in planta (i.e. on leaf surfaces) or in vitro (i.e. on an agar plate), 2) there is overlap between Leaf68 and Leaf257 in the kind of protein functions that both express in planta, and 3) both Leaf68 and Leaf 257 express proteins in each other's presence in planta that they do not express in planta when they colonize the leaf surface by themselves.

We thank the reviewer for the constructive comments.

The authors frame their study in the context of the phenomenon of 'character displacement', which I had not heard of before. The description of how phenotypic plasticity fits into this (lines 52-57) felt a bit too abstract for me, as did the accompanying illustration (Figure 1). I learned more from looking up some of the referenced papers, so I suggest that the authors redo their description and give some concrete examples from the primary literature instead, like this tadpole study:

<https://pubmed.ncbi.nlm.nih.gov/16602305>. It would also help to include one or more relevant studies from the field of microbiology. This would help me and other readers better understand why this paper is different from other papers that have titles like 'Dual transcriptional profiling of xxx and yyy' and that explore interactions between a host and pathogen (e.g. mice and *Toxoplasma gondii*), a host and a beneficial microbe (i.e. wheat and *Azospirillum brasilense*), or as in this paper, two microbes in each other's presence (like two fungi, two bacteria, or a bacterium and a fungus).

We have expanded the description of Figure 1 to improve clarity and added the recommended citation in the introduction. Although the tadpole study is an interesting one, we think it is difficult to elaborate on in a succinct manner because it refers to phenotypic variation in offspring, whereas we study phenotypic plasticity as a result of (reversible) gene expression. We hope that by providing the reference in addition to the ones already included in the original manuscript, we will give the reader a straightforward access to the classic ecological animal studies.

Indeed, dual transcriptional profiling has been used to study interactions, such as host responses to a pathogen and *vice versa*. The reviewer is correct that our study differs from these in that we are comparing two competing species *in situ*. We hope that our description "We examined the interaction of two representative strains of the abundant, leaf-colonizing families *Sphingomonadaceae* and *Rhizobiaceae in planta* under gnotobiotic conditions." makes it sufficiently clear that we are studying two co-occurring species and their dynamic shifts in ecological niche occupancy rather than investing the host response to a bacterium and *vice versa*. We also hope that the title "among a pair of bacterial phyllosphere commensals *in situ*" avoids misleading the reader.

As the authors offered a list of all the proteins that are differentially expressed (e.g. lines 154-171 for Leaf 68), I could not help wondering about the averaging effect of the proteomics approach and how that should guide the authors' interpretation of the data.

For example, if proteins A, B and C were all found to be induced three-fold in the data, does that mean that proteins A, B and C were three-fold induced in all bacterial cells (which is the default assumption here)?

Or could it be that protein A was six-fold induced in 50% of the cells, protein B was 12-fold induced in

another 25% of the cells, and protein C 12-fold induced in the other 25%?

The latter explanation would be consistent with the known environmental heterogeneity along the leaf surface, where some cells find themselves in an environment where for example protein A is important for survival but proteins B and C are not. The current version of the manuscript is completely lacking appreciation for this alternative explanation. Is there a way to get to confirm this, for example by RT-PCR on individual bacterial cells or by using promoter fusions to genes A, B and C?

Indeed, we examined population averages by harvesting and pooling bacteria from 20 plants for one proteomics sample (using a total of four independent replicates) as described in the Materials and Methods section. We agree that there are most likely subpopulations in which bacteria do not encounter each other and therefore do not change their proteome, while other bacteria are in direct competition and respond more strongly than is currently measured at the bulk level. To clarify, we have included a section in the Discussion (l. 470-475) to highlight the future interest to combine spatial information to study bacteria in a heterogeneous environment, here the phyllosphere. Single-cell proteomics of bacteria is technically not possible. Single-cell transcriptomics for bacteria is only in its infancy and currently not applicable to non-model organisms and bacterial communities (see Tang 2021 Nat Methods 18:334, comment to Kuchina et al., 2021 Science 371:798). RT-PCR would target only individual genes and would require the analysis of a large number of bacteria, because transcripts are inherently noisy due to their short half-life. The latter compromises the utility of the approach as the quantification of messenger RNA molecules generally does not always reflect well the proteome (Liu et al., 2016, Cell 165, 535). RT-PCR with single cell resolution would also need to be linked to spatial information, which remains a challenge. Promoter fusions are an interesting option, but they are not applicable at the genome scale and are only suitable for validation of a few candidate genes. With somewhat higher throughput approaches like the one introduced very recently (Dar et al., 2021 Science 373:758) it might be possible to link spatial information with gene expression in the future but is out of the scope of our study. In our study, we chose to validate a gene candidate - identified by our proteomics approach - by site directed mutagenesis, which allowed us to directly demonstrate the importance of dethiobiotin for the interaction. We believe the genetic approach is powerful because it establishes a causal relationship and thus goes beyond gene expression data that transcriptional data can provide.

It would be extremely useful for the interpretation of the data if the authors also provided insight into spatial distribution of Leaf68 and Leaf267 prior to or at the time of sampling.

Do these two strains 'avoid' each other or do they mostly co-habitate? Do they occupy different or overlapping spaces on the leaf? This would be an experiment much like the one done by Monier and Lindow (<https://doi.org/10.1128/AEM.71.9.5484-5493.2005>). Seeing that it might be difficult to get both strains to express a fluorescent protein, perhaps a FISH approach, like the one developed in the same lab would be a suitable alternative.

To directly test whether the bacteria encounter each other in the phyllosphere, we performed microscopy, as suggested. We show that co-localization of the two species does indeed occur and that the two species grow in proximity in the observed time span. In summary, we now provide data to show co-occurrence of the two species at the microscale and added these new data to the manuscript (Supplementary Figure 1; Material and Methods).

The ways in which Leaf68 and Leaf257 are inoculated onto and recovered again from the Arabidopsis phyllosphere are not disclosed until very late in the manuscript (in the Materials and Methods section). For a proper understanding of the results, it is crucial to volunteer this information much earlier in the manuscript. For example, for the proteomics analysis, bacterial cells are retrieved from the leaves by leaf wash and sonication. There is no leaf maceration involved, and so the only cells that are retrieved from the leaves are those that are growing on the leaf surface, not in the leaf interior.

Thank you for pointing this out. We agree that this information should be mentioned earlier in the manuscript. It has now included when we first mention the proteomics approach (l. 126).

Why then do the authors find induction of proteins involved in the catabolism of xylan, which is a component of the plant cell wall, which should not be accessible to bacteria on the leaf surface?

Indeed, xylan is a major component of the plant cell wall, and its accessibility to bacteria on leaf surfaces is not well elucidated. The leaf contains areas where the cuticle thickness can vary greatly (Kosma et al., 2009 Plant Physiol 151:1918) which could lead to xylan accessibility at the junctions where plant cells have to connect and in particular close to veins, leaf margins and at the basis of trichomes. The point was now added to the manuscript (l. 425-427).

Is there another source of xylan on the leaf surface? Is the expression of these proteins truly a response to the presence of xylan, or is there an alternative explanation (e.g. xylose isomerases can interconvert glucose and sucrose; might that be the reason why this protein is produced)? This needs to be addressed in much more detail.

In response to the reviewer's comment, we tested different monosaccharides (including xylose) and found that none of them trigger the secretion of the xylan degrading enzymes (see SDS page of culture supernatants below), in contrast to conditions when cells grow on xylan as sole carbon source. Sequence homology analysis revealed that several of the secreted xylan-degrading enzymes are homologous to enzymes for which high specificity for xylan has been described in other bacteria. Although it is true that xylose isomerase can interconvert glucose and fructose (Kitaoka et al., 1992 Denpun Kagaku 39:281) and fructose is a component of xylan, the bacterium still needs to degrade

xylan to access fructose. However, fructose does not induce secretion of the xylan degrading enzymes.

Inoculation is achieved by pipetting 10 μ L onto Arabidopsis leaves (line 507). This description is incomplete (how many leaves were those 10 μ L distributed over, did each leaf receive more than one drop? all this should be disclosed). The authors should also explain that by inoculating the two strain as a mixture, rather than inoculating one after the other, makes it much more likely for all cells of the two strains to interact with one another and co-habitate the same locations on the leaf then when cells were sprayed one after the other (which allows for some escape from each other, I imagine).

We inoculated each leaf with a drop containing 2 μ L (after 10 days the plant has four leaves) and distributed an additional drop of 2 μ L in the middle of the plant. At the harvest time point, the plant had expanded to stage of four leaves. We added the information to the Material and Method section. We also added a more detailed description to clarify that we inoculated the bacteria as a mixture (see l. 541-543).

The authors do not explain how they differentiated Leaf68 CFUs from Leaf257 CFUs for Figures 2A and 2B.

Thank you for making us aware that this was not evident. We use the natural pigmentation of both strains to differentiate them. *Sphingomonas* Leaf257 forms yellow colonies while *Rhizobium* Leaf68 forms white colonies. We amended the Material and Methods section to specify this (l. 569-571).

Other comments:

Line 144: seeing that the agar plates were also incubated in light, explain why there light-stress proteins induced in leaf-grown cells.

We can only speculate, why this is the case. One reason could be that the agar plates with the bacteria were exposed for a shorter time (3 days instead of 11 days). Another reason could be that light stress is more pronounced in presence of elevated oxygen conditions that might result from plant photosynthesis and/or the production of reactive oxygen species produced by the plant.

Line 212: “>|1.5|” : is this correct? I see what the author try to say here, but |1.5| equals 1.5 so “>|1.5|” becomes “>1.5”, which is incomplete. I suggest to write >1.5 or <-1.5.

The reviewer is correct. To improve clarity, we have implemented the alternative notation as suggested by the reviewer.

Line 339 and throughout: check for typos like this one (When should be We), or line 525: platted should be plated and 'was' should be removed.

Done

Reviewer #2 (Remarks to the Author):

In this manuscript, Hemmerle et al. present a very interesting study of the molecular character displacement observed in two phyllosphere bacterial strains during co-colonization of a leaf. How the observed bacterial diversity in plant-associated habitats is maintained and to what extent microbial interactions contribute to this is a fascinating research area. I think this paper is very well presented and makes an important contribution to the field. I have a few points that I think need further clarification.

We thank the reviewer for the positive feedback.

The authors present compelling evidence that resource utilization and ensuing competition are responsible for the dynamics of the two strains in co-colonization versus mono-colonization. However, I'm still wondering whether some of these dynamics may be driven indirectly by differential responses of the plant host to one or the other strain. For example, in the recent Maier et al. 2021 paper from the Vorholt lab, inoculation with Leaf257 induces considerably more differentially expressed genes in Arabidopsis versus Leaf68. Could this differential response of the host plant also be contributing to the increased abundance of Leaf68 in co-localization? I don't know exactly how one might test this, possibly a co-colonization assay on artificial media as was done for the mono-colonization study.

We appreciate the comment. We cannot exclude that the plant does play a role in this microbe-microbe interaction, although the candidate genes point towards a direct interaction. However, in order to address the reviewer's concern, we tested the interaction using different plant immunity mutants, i.e. bak1/bkk1 (AT4G33430/AT2G13790) (compromised immune signaling downstream of PRRs), npr (AT1G64280) (involved in control of the systemic acquired resistance), jar1 (AT2G46370) (involved in the jasmonic acid signaling), rbohD (AT5G47910) (involved in recognition of bacteria, resulting in ROS-burst and PTI, involved in microbiota homeostasis).

These additional experiments confirmed that the interaction described in our study is robust and occurs also in these immune compromised plant mutants.

The results support the conclusion that the microbial interactions observed in planta are the result of microbe-microbe interaction. We have added the assays on robustness of the interaction in different plant backgrounds to the revised version of the paper (Supplementary Fig. 2 and results I.122-125).

Part of this questioning is coming from the idea that perhaps the dynamic proteome changes observed in these two strains in co-colonization may be fairly agnostic to the particular substrate, so long as the resource profile is such that niche overlap exists. One way the authors might address this is by listing in Supp Table 3A and B which of these proteins are also increased/reduced in the artificial media versus in planta.

We agree and added the information into Supplementary Table 3A + 3B.

Figure 1 and lines 410-411: This is a nice narrative but does it really correspond to the observed results? It almost seems like a non-transitive relationship where Leaf68 is the superior competitor

but due to its auxotrophy can never completely outcompete Leaf257. And Leaf257 has a greater breadth of sugar use which might allow it to avoid extinction. Can this be tested on media? Presumably Leaf68 is the superior competitor for readily available sugars and Leaf257 can coexist because it can degrade alternative energy sources?

These are interesting points. There is no clear competitive advantage of Leaf68 over Leaf257 *in vivo*, both reached similar CFUs/mL. It should be kept in mind that Leaf68 can colonize the plant in mono-colonization (indicating that the plant provides biotin or its precursor). We agree that our results reveal a more complex interaction than the one we used as a starting point. However, Figure 1 was intended to summarize our initial hypothesis and starting point for the study.

The interpretation provided on lines 351-353 suggests that competition is not occurring until Leaf68, via supplementation with Leaf257 dethiobiotin, reaches a certain population size. At which point, Leaf257 switches energy sources at a cost to population growth? Is this a possible hypothesis which can be tested?

It is indeed possible that there are local effects that are directly linked to population size at the microscale. We cannot make claims on the spatial temporal dynamics since we measured a state in which a population "steady-state" was reached. Further insights into the dynamics would be needed that are indeed difficult to acquire. Microfluidics experiments allowing for metabolic exchange might be feasible in the future to study growth of both populations with single cell resolution under fully controlled conditions. We hope the reviewer agrees that such an investigation is beyond the scope of this study.

Minor comments

Line 25-27: Perhaps make it clearer under what conditions these traits are found, mono or co-occurrence.

We agree with the reviewer and specified the conditions to make it clearer.

Line 33: Change when to we?

Done

Line 37: or the evolution of increased competitive ability?

Our intention here was to emphasize the reduction of resource competition through a shift in substrate utilization, rather than the evolution of increased competitive ability. We rephrased the sentence to clarify this.

Line 271: phosphorous?

Changed

Line 440: I don't think the authors demonstrate that dissimilarity increases in co-colonization? Dissimilar in terms of what? In terms of an ordination of the proteins found in each strain under mono versus co-colonization?

We agree and amended the text.

Line 517: So aboveground plant parts were removed from roots?

This is correct. The roots were removed and only the phyllosphere compartment was processed. We added a more detailed description in the Material and Methods (l. 564-565) for clarification.

Line 519-522: From this description it sounds like just the epiphytic bacteria were collected. Please state this clearly somewhere in the main text e.g. line 63.

We used an established washing protocol (Vogel et al., 2012 *Appl Environ Microbiol.* 78:5529) to harvest the bacteria that both were confirmed as epiphytes (see Pfeilmeier et al. 2021 *Nat Microbiol.* 6:852). We now include a short description and refer to the references in addition (l. 109-110, 126, 581-583).

How often do members of these two bacterial families co-occur in aggregates on leaves?

We acquired microscopy images of both bacteria *in planta* showing that they indeed frequently co-localize and form co-aggregates. Exemplary pictures were added to the manuscript (Supplementary Figure 1).

Why would biotin synthase be down regulated in Leaf68 during co-colonization?

We realize that the description in the text was not sufficiently clear. Briefly, the biotin synthase BioB is unstable and oxygen sensitive. In addition, the enzyme is consumed to a considerable extent during the reaction (thus acting like a substrate), as a fraction of the protein is sacrificed during the reaction of dethiobiotin to biotin and subsequently degraded (Choi-Rhee et al. 2005 *Chem Biol.* 12:461). Consistent with Choi-Rhee et al., we hypothesize that the reduction of BioB during co-colonization is a consequence of higher dethiobiotin availability by Leaf257 and higher dethiobiotin turnover by Leaf68, which may lead to proteolytic destruction of BioB that can be seen as a reduction in the proteomics data. We have amended the text and refer to the studies mentioned above.

Figure 3: In A, it's a bit funny to have error bars extend above 100%. Possibly overlay a strip chart to show individual data points?

We appreciate the comment and agree. We changed Figure 3A accordingly.

Reviewer #3 (Remarks to the Author):

In this paper, Hemmerle et al. use proteomics and growth experiments in vitro and in planta to investigate the metabolic interaction between two common phyllosphere strains. They show that these two strains alter their gene expression patterns plastically when colonising the plant together, compared to when they are inoculated alone. They dig into a few particular proteins and find a facilitative interaction related to the production of the vitamin biotin. They also show metabolic shifts in one species towards the uptake of xylan when the partner is present. Overall, I find this approach to be very powerful, and is a rare opportunity to look into how two microbes interact in a setting that is very close to the natural scenario. The experiments are mostly very complete (see below) and the paper is very well written.

We thank the reviewer for the positive feedback.

A few things were not very clear to me when reading the paper, though, which I discuss next. I also list a number of general questions below that would be good to address in the introduction or the discussion sections of the paper, and some minor comments. My main recommendation for improvement is to better outline what the authors think is happening with the biotin interaction. By rereading the text and looking at supplementary Fig. 6A-C (I think it would be helpful to put the message of these growth curves in the main text), I understood that the authors are proposing the following scenario: Leaf257 makes more dethiobiotin available, which Leaf68 wt can make use of to make more biotin, but the Leaf68 *bioB* mutant cannot. The wt then “steals” dethiobiotin from Leaf257, presumably because it can take it up faster, leading to a competitive advantage and facilitation compared to when it’s alone.

The reviewer is correct that Leaf257 makes more dethiobiotin available, which Leaf68 can use resulting in an increasing its abundance. The underlying principle is that for the auxotroph Leaf68 the dethiobiotin is growth limiting and an increased pool will lead to higher colonization (as we show with the supplementation of the pure chemical in the revised manuscript in addition to the biotin supplementation). The secretion of dethiobiotin by Leaf257 could be mainly mediated through passive leakage (Pfiffeteau and Gaudry 1985 Biochim Biophys. 816:77). Leakage of metabolites is a general principle observed with various different metabolites and in different biological systems (Pinu et al., 2018 Metabolomics 14:43; Douglas 2020 Trans R Soc Lond B Biol Sci. 375:20190250; Raatz et al., 2018 Limnol Oceanogr Methods 16:629). We expanded the manuscript to better highlight this point more clearly (l. 357-361, 388-393, 446-447).

If this is the case, then wouldn’t it make sense to supplement with dethiobiotin rather than biotin in Fig. 5C-D?

To address this point, we repeated the experiment with dethiobiotin (and biotin as a control). The results show that dethiobiotin supplementation results in a significant increase of Leaf68 wild type but not in the *bioB* mutant. Furthermore, the Leaf68 *bioB* mutant did not change its abundance upon dethiobiotin treatment during mono- and co-colonization, but significantly increased its abundance in both colonization conditions when treated with biotin. Thus underlining that the interaction between Leaf68 and Leaf257 is facilitated via the biotin precursor dethiobiotin. We adjusted Figure 5 to include the new data and combined the panels as suggested by the reviewer (Figure 5C).

The authors explain that “The addition of biotin significantly increased the abundance of Rhizobium Leaf68 wild type during mono colonization of the plant (Figure 5D) and to similar extent as during co-colonization with Sphingomonas Leaf257” (it’s mentioned a second time elsewhere “the finding that the addition of biotin to plants increases the population of Rhizobium Leaf68”). However, I don’t really see this in Fig. 5D: if I compare the first and the 5th bar, they look similar and there is no statistical test comparing them.

Thank you for pointing this out. We realize that this needs an additional graph showing that in mono-association Leaf68 wild type colonizes the plant better when either dethiobiotin or biotin are added

to the phyllosphere, while the Leaf68 *bioB* mutant only colonizes better through addition of biotin but not dethiobiotin. Moreover, Leaf68 reaches a maximum during the interaction with Leaf257, regardless of the treatment with biotin or dethiobiotin. The *bioB* mutant does not benefit from presence of Leaf257, but the interaction could be rescued by the addition of biotin but not dethiobiotin. Leaf257 is not affected by the treatments during mono-colonization. We have added this comparison into the Supplementary Figure 8A and B.

Finally to this point, why does *Sphingomonas* not use the dethiobiotin it produces? Isn't it surprising that it's freely available in excess?

There are several studies showing that different metabolites are secreted by bacteria and can be indicative for the metabolic state of the cell (Granucci et al., 2015 *Mol Biosyst.* 11:3297; Pinu et al., 2018 *Metabolomics* 14:43). For vitamins the secretion of pantothenate was previously described (Jackowski, Rock 1981 *J Bacteriol.* 148:926). In case of biotin, it was described for *E. coli* that the efflux of biotin is mainly mediated by diffusion mechanisms (Pfiffeteau and Gaudry 1985 *Biochim Biophys.* 816:77), suggesting that the "free" pool is indeed based on leakage of biotin or its precursor dethiobiotin. We amended the manuscript for clarification (l. 446-447).

In short, I am a bit confused about this part and suggest the authors clarify the text a bit and apply the relevant statistical comparisons. If I understood correctly, then I would also recommend them repeating the experiment with the biotin addition but with dethiobiotin.

We added the appropriate statistical tests and repeated the experiment with biotin and dethiobiotin addition (see Figure 5C). We also added a description (l. 446) into the discussion in regard to the secretion of biotin.

My other point is regarding the narrative of the paper that the main result is one of plastic niche separation. I agree that there is a clear shift by Leaf257 to metabolise xylan, but from the introduction, I was expecting to see more evidence for niche separation in the genes that are down-regulated. Is there perhaps more to analyse there? Instead, the paper focuses a lot on the biotin interaction, which seems to be less about niche separation and more about exploitation of the biotin precursor. This is all fine, of course. I am just wondering whether the authors may want to reconsider the narrative or make it more nuanced given the interesting results concerning the biotin interaction.

We see the point and indeed the initial hypothesis on plastic niche separation was validated by xylan metabolization, while the discovery on niche facilitation was not anticipated. We work towards converging both aspects already in the abstract by highlighting the cross species facilitation and the conclusion sentence "Our results show that dynamic character displacement and niche facilitation mediated by phenotypic plasticity can contribute to species coexistence." We are open to further advice to improve the storyline.

Finally, some more general questions/comments:

- Do these two isolates come from the same plant? Would the studied interaction have co-evolved?

Thank you for raising this question. No, *Rhizobium* Leaf68 and *Sphingomonas* Leaf257 were not isolated from the same plant. Therefore, we do not know whether this species interaction was shaped by co-evolution. However, *Sphingomonas* Leaf32 and *Rhizobium* Leaf68 come from the same plant, and we observed a similar interaction based of CFUs per g FW *in planta*. This finding may indicate a general co-evolution of *Rhizobium* and *Sphingomonas* spp. but would require further testing and we find it somewhat preliminary at this stage. We now refer to future work in the discussion section to include this line of thought.

- Are the agar samples mono- and co-colonization as well?

The agar samples are only mono-colonization. We amended the corresponding sentences in the abstract, results (l. 134 and Material and Methods section (l. 589) to clarify this point.

- Some statistical tests are missing: e.g. Fig. 4B,C.

We added the appropriate statistical tests.

According to the text, in Fig. 5B a statistical comparison is needed between Leaf68bioB and Leaf68bioB+Leaf257. Related: it's a bit awkward to talk about "a complete loss of the interaction" l. 350 in a statistical sense

Thanks for pointing this out, we agree. The statistical analysis for the comparison was performed and added to Supplementary Figure 8A. The sentence was amended it now reads "...we observed a significant loss of the interaction with the Rhizobium Leaf68 *bioB* mutant (Figure 5B)."

And a few questions from someone who is not an expert in proteomic analyses:

- How do you remove plant proteins from analysis? How do you ensure that plant responds the same to mono- and co-colonization?

To investigate the degree of plant contamination, we include the plant proteome in the database search and the corresponding hits were then filtered. We did detect peptides originated from plant proteins but the majority (more than 84%) of peptides measured are from the bacteria. Note that our protocol was a harsh washing protocol rather than macerating the plant cells that indeed would have made our proteomics approach unfeasible. We amended the Material and Methods section for clarification (l. 582, 644-647).

Regarding the second question, we cannot exclude that the plant might respond differently during mono- and co-colonization. We now demonstrated that the interaction we observe is robust when we used plant immunity mutants (Supplementary Figure 2) (see also our response to reviewer 2).

- Do you correct for population sizes and changes in these when looking at fold-changes?

We realize that the description in the text regarding the normalization for the proteomics approach was not clear and we have adjusted the text accordingly. Briefly, we normalize for each strain independently over all precursor ions detected. E.g. when we look at the Leaf257 proteome we only use precursor ions specific for Leaf257 to normalize the data sets obtained from mono- and co-colonization. Therefore, this approach will normalize over population size changes sufficiently. Note, that the proteomes for co-colonization were compared to the "artificial mixture". This "artificial mixture" resembles the scenario of both bacteria colonizing the plant without interacting with each other. To generate the "artificial mixture" the peptides obtained from the digested protein samples of mono-colonized plants by either Leaf257 or Leaf68 were mixed at the same concentrations. This provides an additional control on sample complexity and a reference condition for the label-free quantification approach (l. 643-646).

Minor comments:

l. 64 typo: import -> important

Done

l. 116-119: labelling of Figures 2A, B not quite right

Changed

Fig. 2: what is leaf68mix vs. artmix? Specify in caption

We thank the reviewer for pointing this out. We added a full description into the Figure 2.

l. 275: increased utilization of xylan compared to what? Mono-colonisation?

For further clarification, we added a description in l. 287.

Fig. 3A: what is Mix and ArtMix?

We appreciate the comment and agree that the description is not that clear. We changed the Figure 3A and labeled the mixture as co-colonization and artificial mixture as mono-colonization.

l. 339: When -> We

Changed

l. 354: in respect -> with respect

Changed

Fig. 5C, D: I think it's best to combine these panels into 1, it's a bit strange that the data are repeated in both panels

Thank you for this helpful feedback. As the reviewer suggested we combined both panels into one and adjusted Figure 5 accordingly (see comment above and Figure 5).

l. 373ff: I don't see the comparison you mention in Fig. 5D, there are no statistics for mono-colonisation in H₂O versus biotin

The reviewer rightfully pointed out that this comparison is missing. We added this to Supplementary Figure 8B.

l. 316: "All[..] mutants grew similarly compared to the wild type during growth on minimal medium with glucose" but Sup. Fig. 6 shows that the *phoB* mutant grows quite a bit worse

We want to thank the reviewer for pointing this out. We corrected this in the text accordingly.

REVIEWERS' COMMENTS

Reviewer #1 (Remarks to the Author):

My concerns have been addressed in this revised version of the manuscript.

Reviewer #2 (Remarks to the Author):

Well done addressing all reviewer comments, I have no additional comments.

Reviewer #3 (Remarks to the Author):

I would like to thank the authors for the thorough revision of the manuscript. The additional experiments make the story more complete from my perspective and the logic has been clarified in the text.